# Nonlinear Sequence Data Embedding by Monotone Variational Inequality

**Jonathan Y. Zhou, Yao Xie**
School of Industrial & Systems Engineering
Georgia Institute of Technology
Atlanta, GA 30332
`jyz@gatech.edu, yao.xie@isye.gatech.edu`

## Abstract

In the wild, we often encounter collections of sequential data such as electrocardiograms, motion capture, genomes, and natural language, and sequences may be multichannel or symbolic with nonlinear dynamics. We introduce a method to learn low-dimensional representations of nonlinear sequence and time-series data without supervision which has provable recovery guarantees. The learned representation can be used for downstream machine-learning tasks such as clustering and classification. The method assumes that the observed sequences arise from a common domain, with each sequence following its own autoregressive model, and these models are related through low-rank regularization. We cast the problem as a convex matrix parameter recovery problem using monotone Variational Inequalities (VIs) and encode the common domain assumption via low-rank constraint across the learned representations, which can learn a subspace approximately spanning the entire domain as well as faithful representations for the dynamics of each individual sequence incorporating the domain information in totality. We show the competitive performance of our method on real-world time-series data with baselines and demonstrate its effectiveness for symbolic text modeling and RNA sequence clustering.

## 1 Introduction

Collections of time-series data, where each sequence is represented by a series of points indexed over time, are ubiquitous and increasingly prevalent. Notable examples include physiological signals (Cohen, 2014; Alday et al., 2020), power systems (Van Wijk & Van Selow, 1999), financial data (Tsay, 2005; Min et al., 2021), computer networks (Basu et al., 1996), and electronic health records (Reyna et al., 2020; Rasmy et al., 2021). In addition to traditional time series data, other sequential data like gene and protein sequences (Argelaguet et al., 2020; Jumper et al., 2021) as well as natural language have garnered significant attention, particularly with the advent of large language models (Brown et al., 1992; Peters et al., 2018; Reimers & Gurevych, 2019; Cer et al., 2018).

Learning high-quality representations of sequences and time series (Mikolov et al., 2013) is an essential building block for understanding the dynamics underlying observed sequences, enabling informed decision making and downstream machine learning tasks (Trirat et al., 2024). A key paradigm underlying the unsupervised representation learning has been that of self-supervised learning (Shwartz Ziv & LeCun, 2024), where we first solve some auxiliary task (e.g., autoregression or reconstruction), which leads implicitly to a compressed representation of the input data (Murphy, 2023; Kingma & Welling, 2022). The development of self-supervised methods for natural language has been, in turn, paralleled by embedding methods for other types of sequential data, with there now being a burgeoning literature on time series and sequence representation (Lafabregue et al., 2022; Krishnan et al., 2022).

A lasting challenge in bringing representation learning to time series is how to learn the information *common to the entire domain* in conjunction with faithful *individual* representations of each sequence. Indeed, when learning a model for time series or sequence data a common assumption to make is that the sequence observations arise from repeated realizations of a single random source. While

this is effective in the natural language setting where there exists a shared "universal" embedding space language (Yang et al., 2020) backed up by a large amount of available training data, many time series data are often highly domain-specific in the sense that each domain is distinct from another (e.g., Electrocardiogram (ECG) vs power systems). Sequences may be also highly distinct from one another (e.g., healthy vs sick patients) and our observations of each individual be partial or limited. To this end, recent empirical evidence indicates (Tan et al., 2024) that augmenting time-series prediction with large language models results in performance no better than models trained from scratch, and removing the LLM components. For many settings, the individual processes which we receive observations for may in fact also be *substantially different among themselves* (e.g., differences between sick and healthy patients). There thus is a challenge in balancing learning the common dynamics of a set of observed sequences in addition to faithfully representing the dynamics of each sequence. This is especially the case when observations of all sequences individually are *limited or otherwise partially observed*. To this end, we take inspiration from an area where the above challenges are common and well known — low-rank matrix recovery (Davenport & Romberg, 2016; Candes & Tao, 2010; Candes & Plan, 2010; Ahmed & Romberg, 2015; Juditsky & Nemirovski, 2020) — previously applied to collaborative filtering problems and develop it towards the general sequential and time series representation learning setting, enabling us to bring provable recovery guarantees to the modeling of a broad class of sequences with autoregressive character.

To this end, we introduce an approach for unsupervised learning of low-dimensional representations for collections of nonlinear sequences, time-series, and dynamical systems based on the assumption that each sequence behaves according to its own autoregressive model but that the sequences are related to each other through low-rank regularization. We cast the problem as a computationally efficient convex matrix parameter recovery problem using monotone VIs. This formulation maintains problem convexity and recovery guarantees, while allowing for a broad range of autoregressive sequence dynamics through an arbitrary monotone link function. By enforcing a low-rank assumption across the learned representations we efficiently learn a subspace to capture entire domain. We apply our method to real-world time-series data and demonstrate its effectiveness in symbolic text modeling and RNA sequence clustering. On many datasets, our method performs comparably to neural-network based deep representation models.

## 1.1 Related work

We review related work in time-series representation learning, clustering, and classification. Simple methods include feature extraction (Ye & Keogh, 2009) or defining a distance metric between time-series (Cormen et al., 2001; Müller, 2007; Bagnall et al., 2017). Another approach is to model each series, which aligns with our model-based representation approach (Smyth, 1996; Kalpakis et al., 2001). Recent time-series representation learning methods often use contrastive learning to distinguish sequences, employing deep networks to treat sub-samples of the same sequence as positives and different sequences as negatives (Yang & Hong, 2022; Yue et al., 2022; Xiao et al., 2024; Fraikin et al., 2024; Wang et al., 2023). These approaches focus on neural architecture, data augmentation for robustness, and contrastive learning strategies (Ma et al., 2019; Fortuin et al., 2020; Devlin et al., 2019).

In our work, we adopt autoregression as the auxiliary task. Unlike techniques which use contrastive learning to indirectly learn an encoder for the latent space, we do not assume inherent similarities or differences across sequences. Instead, we explicitly constrain the representations to lie in a low-rank space. We motivate our work from the perspective of *low-rank matrix recovery* (Davenport & Romberg, 2016), common in other areas of machine learning and serving as the foundation for principal component analysis (Hotelling, 1933), classical methods in natural language processing (topic modeling) (Blei et al., 2003; Blei, 2012) and collaborative filtering (Koren et al., 2009). Problems in this area typically admit *convex formulations* and come with provable recovery guarantees (Juditsky & Nemirovski, 2020). Most recently, a line of work has on *signal recovery by convex optimization* has loosened the structural assumptions needed for signal (time-series) recovery in an autoregressive context while still maintaining problem convexity, using VIs with monotone operators the main tool (Juditsky et al., 2023; 2020; Juditsky & Nemirovski, 2019).

The basic idea that sequences (time-series) can be represented in a low-dimensional space (e.g., by latent factors) has a long history, such as hierarchical time-series models (Laird & Ware, 1982; Gamerman & Migon, 1993). More recently, Kirchmeyer et al. (2022) and Kostic et al. (2024) address

dynamical systems learning, where either a context or latent vector aids in governing each sequence's dynamics, in contrast to directly learning an encoder from observations into a latent space.

## 2 PROBLEM SETUP

We aim to represent observations into $N$ vector-valued time series of length $T$ each of the form $\{\mathbf{x}_{i,t}\}$, where $\mathbf{x} \in \mathbb{R}^C$, $t \in [T]$, and $i \in [N]$. Here, we introduce the algorithm under the assumption that all sequences have the same length for notational clarity. However, the proposed results and algorithms can be extended to accommodate sequences that are partially observed or of varying lengths.

The sequences are sampled from a common domain independently of each other across $i$, but have temporal dependence across $t$. We refer to the history of events for sequence $i$ up to time-point $t$ as $\mathcal{H}_{i,t} := \{\mathbf{x}_{i,s} \mid s < t\}$. We expect the behavior at event $\mathbf{x}_{i,t}$ to be a function of past observations. Namely, at each time-point we suppose $\mathbf{x}_{i,t}$'s dependence on its past values $\mathcal{H}_{i,t}$ is sufficiently captured by a nonlinear vector autoregressive model of order $d$ with $C$ channels. In particular, we package the preceding $d$ observations with a bias term as a vector

$$\boldsymbol{\xi}_{i,t} = \mathrm{vec}(1, \{\mathbf{x}_{i,t-s}\}_{s=1}^d) \in \mathbb{R}^{Cd+1},$$

where $\mathrm{vec}(\cdot)$ arranges its arguments into a single column vector so that

$$\mathbb{E}[\mathbf{x}_{i,t} \mid \mathcal{H}_{i,t}] = \phi(\mathbf{R}_i \boldsymbol{\xi}_{i,t}), \quad \forall i \in [N]. \tag{1}$$

Note that we allow each sequence $i \in [N]$ to *have its own own model*. In particular, we aim to learn matrices $\mathbf{R}_i \in \mathbb{R}^{C \times (Cd+1)}$ which serve as model parameters for prediction of the focal observation $\mathbf{x}_{i,t}$. For each $\mathbf{R}_i$, we write $\mathbf{b}_i = \mathrm{vec}(\mathbf{R}_i) \in \mathbb{R}^m, m := C^2 d + C$ as the corresponding parameter vector sufficient to capture the dynamics to the $i^{\text{th}}$ sequence.

The structure of our model is inspired by Generalized Linear Models (GLMs) (Nelder & Wedderburn, 1972), with a monotone link function $\phi : \mathbb{R}^C \to \mathbb{R}^C$ that captures sequence dynamics. The choice of $\phi$ naturally corresponds to a variety of different models and phenomena, for example:

**Vector auto-regression:** $\phi(\mathbf{x}) = \mathbf{x}; \mathbf{x} \in \mathbb{R}^C$, e.g. motion capture, Electrocardiogram (ECG) signals.

**Symbolic sequences:** $\phi(\mathbf{x}) = \exp(\mathbf{x})/\sum_i \exp(x_i); \mathbf{x} \in [\Sigma]^C$, e.g. natural language, genes.

**Count processes:** $\phi(\mathbf{x}) = \exp(\mathbf{x}); \mathbf{x} \in \mathbb{Z}_{\geq 0}^C$, e.g. traffic intensity, call center arrival rates.

**Bernoulli Processes:** $\phi(\mathbf{x}) = \exp(\mathbf{x})/(1 + \exp(\mathbf{x})); \mathbf{x} \in \mathbb{B}^C$, e.g. wildfire presence, neuron firing.

We do not restrict the mechanics of the link function $\phi$ beyond the fact it is monotonically increasing. We remark also that each vector $\mathbf{b}_i$ corresponding to each sequence may itself be high dimensional.

The key aspect of our method for low dimensional representation learning lies in the common domain assumption, which should limit how the sequences are similar (different) through their individual model. We leverage this information by a *low rank* assumption on the space of parameters by which each sequence is described. In this way, we constrain the individual $\mathbf{b}_i$ to lie approximately on a *low dimensional linear subspace* of the possible full parameter space $\mathbb{R}^m$. The representation of each sequence's parameter within this subspace may be taken as a low-dimensional embedding and used for downstream tasks such as clustering, classification, and anomaly detection.

In particular, we consider the autoregressive sequence model introduced in (1), allowing $\mathbf{b}_i$ to be those parameters unique to the $i^{\text{th}}$ sequence. Allow the matrix $\mathbf{B} = [\mathbf{b}_1 \quad \ldots \quad \mathbf{b}_N] \in \mathbb{R}^{m \times N}$ denote the parameters across all the sequences. We aim to recover a good choice of the matrix $\mathbf{B}$ without supervision and balancing two goals: (1) we desire each $\mathbf{b}_i$ to be as faithful to the generating dynamics of their respective observed data as possible; (2) we hope to leverage the *common domain assumption* about the sequences and use information from the other sequences to inform the prediction of the focal sequence. To express the corresponding low-rank constraint, consider the rank $r$ Singular Value Decomposition (SVD) of $\mathbf{B}$

$$\mathbf{B} = \mathbf{U}\boldsymbol{\Sigma}\mathbf{V}^* = \sum_{k=1}^r \sigma_k \mathbf{u}_k \mathbf{v}_k^* \tag{2}$$

where $\boldsymbol{\Sigma} = \mathrm{diag}\{\sigma_k\}_{k=1}^r$ corresponds to the singular values, columns of $\mathbf{U} = [\mathbf{u}_k]_{k=1}^r \in \mathbb{R}^{m \times r}$ form an orthobasis in $\mathbb{R}^d$, and columns of $\mathbf{V}^* = [\mathbf{v}_k]_{k=1}^r \in \mathbb{R}^{r \times N}$ form an orthobasis in $\mathbb{R}^N$. The

recovered columns $\mathbf{C} := \mathbf{\Sigma V}^* = [\mathbf{c}_i]_{i=1}^N \in \mathbb{R}^{r \times N}$ give an $r$-dimensional representation for each of the $N$ sequences. Likewise, the subspace $\mathrm{sp}\{\mathbf{u}_k\}_{k=1}^r$ describes the *common domain* from which the generating processes of the sequences arise. We consider the low dimensional representation $\mathbf{c}_i$ for the $i^{\text{th}}$ sequence $\mathbf{b}_i = \mathbf{U}\mathbf{c}_i$ as an embedding of the dynamics for the $i^{\text{th}}$ sequence.

Because rank-constrained optimization is in general an NP-hard problem (Natarajan, 1995), to enforce the low-rank requirement on $\mathbf{B}$, we instead constrain our setup to a *nuclear norm ball*. The nuclear norm is given by $\|\mathbf{X}\|_* = \sum_{j=1}^r \sigma_i(\mathbf{X})$ where $\sigma_i$ is the $i^{\text{th}}$ singular value of the matrix $\mathbf{X}$. The nuclear norm is the tightest convex relaxation to matrix rank (Recht et al., 2010) leading to tractable parameter recovery and allows us to leverage a long line of work from convex optimization and matrix recovery (Cai et al., 2010; Davenport & Romberg, 2016; Nesterov & Nemirovski, 2013).

We model each sequence as originating from an individual stochastic source, evolving according to the parametric observation model of (1). This model, detailed in Juditsky & Nemirovski (2020), balances the need for learned dynamics to closely resemble the original time-series observations (by using the most general convex model) with the requirement for efficient and identifiable parameter recovery via first-order methods (Facchinei & Pang, 2003). The choice to make the observation model convex is motivated not only by parameter recovery guarantees but also by the need to ensure regularity of the parameter space for low-rank estimation of sequence parameters in aggregate. By framing time-series and sequence representation learning as a convex low-rank matrix parameter estimation task, our method is particularly well-suited for representation learning in contexts with limited, partially observed, and highly heterogeneous sequence data.

## 3 METHOD

In the following, we present our method, first for linear auto-regressive models and then for general non-linear auto-regressive models, including categorical sequences.

### 3.1 LOW RANK TIME-SERIES EMBEDDING FOR LINEAR AUTO-REGRESSIVE MODELS

First, suppose events $\mathbf{x}_{i,t} \in \mathbb{R}^C$ obey a linear autoregressive model corresponding to (1) with $\phi(\mathbf{x}) = \mathbf{x}$. To recover the parameter matrix $\mathbf{B} = [\mathrm{vec}(\mathbf{R}_i)]_{i=1}^N \in \mathbb{R}^{m \times N}$, a natural choice is to take least squares loss and write

$$\min_{\mathbf{B} \in \mathbb{R}^{d \times N}} \frac{1}{N} \sum_{i=1}^N \left( \frac{1}{T-d} \sum_{t=d+1}^T \|\mathbf{x}_{i,t} - \phi(\mathbf{R}_i \boldsymbol{\xi}_{i,t})\|_2^2 \right) \qquad \text{s.t.} \qquad \|\mathbf{B}\|_* \le \lambda. \qquad (3)$$

where $\lambda \ge 0$ is a regularization parameter governing the rank of $\mathbf{B}$.

**Low-rank recovery and nuclear norm regularization.** We now discuss Program (3) in the context of low-rank matrix recovery (Davenport & Romberg, 2016). We aim to recover matrix $\mathbf{B}$, but instead of observing it directly, we receive it through indirect observations through successive linear *measurement operators*

$$\mathcal{A}_t(\mathbf{B}) := \mathrm{vec}([\mathbf{R}_i \boldsymbol{\xi}_{i,t}]_{i=1}^N) : \mathbb{R}^{m \times N} \to \mathbb{R}^{CN} \qquad (4)$$

which at each time-step $t$, provide for the prediction of the states $\mathbf{x}_{i,t}$ across all observed sequences $i \in [N]$. We likewise define $\mathbf{y}_t := \mathrm{vec}([\mathbf{x}_{i,t}]_{i=1}^N) \in \mathbb{R}^{CN}$ to be the true values for the focal time-point across all $N$ sequences. We then use all of the observed temporal slices of size $d+1$ running up to time $T$, to and consider the least squares loss via the program

$$\min_{\mathbf{B}} \hat{\ell}(\mathbf{B}) := \frac{1}{N(T-d)} \sum_{t=d+1}^T \|\mathcal{A}_t(\mathbf{B}) - \mathbf{y}_t\|_2^2 \qquad \text{s.t.} \qquad \|\mathbf{B}\|_* \le \lambda, \qquad (5)$$

which is a Lipschitz smooth convex program on the nuclear ball of radius $\lambda$ (Shapiro et al., 2021). Program (5) is exactly the same as Program (3) except placed in a matrix recovery context. We aim to recover the optimal $\mathbf{B}$ from the samples while accounting for the global structure. When $\lambda$ is arbitrarily large, there is no constraint on $\mathbf{B}$ and Program (5) corresponds to fitting each sequence individually with no global information. On the other extreme, forcing $\mathbf{B}$ to be rank one constrains

the models of each sequence to be multiples of each other. Intermediate values of $\lambda$ correspond to various trade-offs between learning common global structure and personalization to individual sequences.

Program (5) can be readily cast as a Semidefinite Program (SDP) solvable using standard interior point methods (Ben-Tal & Nemirovski, 2001). However, as the size of $\mathbf{B}$ may reach into the hundreds of thousands of decision variables, we turn our discussion to efficient first-order algorithms (Combettes & Pesquet, 2011; Parikh & Boyd, 2014) analogous to those for linear inverse problems (Beck & Teboulle, 2009). Indeed, consider the following Proximal Gradient (PG) procedure

$$\mathbf{B}_{k+1} = \text{Prox}_{\gamma_k \delta_\lambda}(\gamma_k \nabla_{\mathbf{B}_k}[\ell(\mathbf{B}_k)]), \qquad \mathbf{B}_0 \in \{\mathbf{X} \mid \|\mathbf{X}\|_* \leq \lambda\}, \tag{6}$$

consisting of a gradient step followed by proximal projection for an appropriately chosen sequence of steps $\{\gamma_k\}$. In this case, the *prox-mapping* associated with the indicator of the nuclear ball with radius $\lambda$ is the projection

$$\text{Prox}_{\gamma_k \delta_\lambda}(\mathbf{X}) = \underset{\mathbf{Y} \in \{\mathbf{Y} \mid \|\mathbf{Y}\|_* \leq \lambda\}}{\arg\min} \|\mathbf{X} - \mathbf{Y}\|_2.$$

We can compute this projection mapping using Singular Value Thresholding (SVT), eliminating the small singular values (Cai et al., 2010; Nesterov & Nemirovski, 2013). Namely, with $\mathbf{X} \in \mathbb{R}^{m \times n}$, we first compute its SVD $\mathbf{X} = \mathbf{U} \text{diag}\{\sigma_i\}_{i=1}^m \mathbf{V}^*$. We then project the singular values onto a simplex

$$\mathbf{t} = \min_{\mathbf{t} \in \mathbb{R}^m} \left\{ \sum_{i=1}^m (\sigma_i - t_i)^2 \mid \sum_{i=1}^m t_i = \lambda, \mathbf{t} \geq \mathbf{0} \right\}. \tag{7}$$

The matrix projection can then be constructed by reapplying the singular vectors

$$\text{Prox}_{\gamma_k \delta_\lambda}(\mathbf{X}) = \mathbf{U} \text{diag}(\mathbf{t}) \mathbf{V}^*.$$

In the worst case, Program (7) requires that we find $\mathcal{O}(m)$ singular values. However, it is typically the case that the parameter matrix $\mathbf{B}$ has a decaying spectrum of singular values, and we take a small choice of $\lambda$ in (6) as to ensure the representation is low rank. Thus, we typically need only to calculate a few largest singular values of $\mathbf{B}$ need, for instance via the Golub-Kahan-Lanczos bidiagonalization processes (Golub & Van Loan, 1996). We compute singular values and vectors in an incremental fashion, up to the $r^{\text{th}}$ singular value that still satisfies $\sigma_r > \frac{1}{r}(-\lambda + \sum_{j=1}^r \sigma_j)$ (Duchi et al., 2008). Then, the optimal solution to (7) is given in closed form by

$$\mathbf{t} = \begin{cases} \sigma_i - \frac{1}{r}\left(-\lambda + \sum_{j=1}^r \sigma_j\right) & i \leq r \\ 0 & i > r \end{cases}.$$

In our real-data experiments, the iteration cost is typically dominated by the cost of computing the gradient, which can be mitigated by stochastic approximation.

### 3.2 NONLINEAR TIME-SERIES EMBEDDING BY MONOTONE VI

We now extend our discussion to the nonlinear case. Our goal is to likewise form a rank-constrained stochastic estimate to $\mathbf{B}$ when we recieve observations according to (1).However, with arbitrary monotone link $\phi : \mathbb{R}^C \to \mathbb{R}^C$ the Least Squares (LS) approach outlined in (3) and (5) loses convexity and computational tractability in general. Likewise, parameter estimation by Maximum Likelihood Estimation (MLE) also becomes computationally difficult (Juditsky & Nemirovski, 2020). By contrast, we shall cast the parameter recovery problem into a monotone VI formulation, the most general type of convex program with known methods to efficiently find high accuracy solutions (Juditsky et al., 2023; Juditsky & Nemirovski, 2019; Juditsky et al., 2020).

**Preliminaries on monotone VI.** A *monotone vector field* on $\mathbb{R}^m$ with modulus of convexity $\beta$ is a vector field $G : \mathbb{R}^m \to \mathbb{R}^m$ such that

$$\langle G(\mathbf{x}) - G(\mathbf{x}'), \mathbf{x} - \mathbf{x}' \rangle \geq \beta \|\mathbf{x} - \mathbf{x}'\| \qquad \forall \mathbf{x}, \mathbf{x}' \in \mathcal{X}$$

when $\beta > 0$, $G$ is *strongly monotone*. For some convex compact set $\mathcal{X} \subseteq \mathbb{R}^m$, a point $\mathbf{x}^*$ is a *weak solution* to the VI associated with $(G, \mathcal{X})$ if for all $\mathbf{x} \in \mathcal{X}$ we have $\langle G(\mathbf{x}), \mathbf{x} - \mathbf{x}^* \rangle \geq 0$. If $G$ is strongly monotone and a weak solution exists, then the solution is unique. When $\langle G(\mathbf{x}^*), \mathbf{x} - \mathbf{x}^* \rangle \geq 0$ for all $\mathbf{x} \in \mathcal{X}$, we term $\mathbf{x}^*$ a *strong solution* to the VI. When $G$ is continuous on $\mathcal{X}$, all strong solutions are weak solutions and vice versa (Facchinei & Pang, 2003).

**Monotone VI for nonlinear parameter recovery.** We turn our attention now to the construction of a Monotone VI which has as its root optimal parameters corresponding to Model (1). We will use the same operator $\mathcal{A}_t$ from the linear case defined in (4) together with its corresponding adjoint $\mathcal{A}_t^* : \mathbb{R}^{CN} \to \mathbb{R}^{m \times N}$ which takes the pre-image of the multichannel predictions (observations) $\mathbf{y_t} = \text{vec}([\mathbf{x}_{i,t}]_{i=1}^N)$ and maps them back to the parameter space. Note that $\mathcal{A}_t^*$ can be computed using the below formula

$$\mathcal{A}_t^*(\mathbf{y}) = [\text{vec}([x_{i,c}\mathbf{R}_i^T\mathbf{e}_c])_{c=1}^C)]_{i=1}^N : \mathbb{R}^{CN} \to \mathbb{R}^{m \times N}$$

where $\mathbf{e}_c$ is $c^{\text{th}}$ standard basis. The adjoint, for each entry and channel, multiplies the parameters by the value of the observation corresponding to the channel.

Consider now the vector field on the space of matrices

$$
\begin{aligned}
\Psi(\mathbf{B}) &= \frac{1}{N}\mathbb{E}_t[\mathcal{A}_t^*(\phi(\mathcal{A}_t(\widehat{\mathbf{B}})) - \mathbf{y})] : \mathbb{R}^{(C^2d+C)\times N} \to \mathbb{R}^{(C^2d+C)\times N} \\
&= \frac{1}{N(T-d)}\sum_{t=d+1}^{T}[\mathcal{A}_t^*(\phi(\mathcal{A}_t(\mathbf{B}))) - \mathcal{A}_t^*(y_t)] \\
&= \frac{1}{N(T-d)}\sum_{t=d+1}^{T}\mathcal{A}_t^*[\phi(\mathcal{A}_t(\mathbf{B})) - \mathbf{y}_t].
\end{aligned}
\tag{8}
$$

where we extend the link function $\phi$ to act sample wise. Notice that the matrix $\mathbf{B}$ of true generating parameters is a zero of $\Psi$,

$$
\begin{aligned}
\Psi(\mathbf{B}) &= \mathbb{E}_t[\mathcal{A}_t^*(\phi(\mathcal{A}_t(\mathbf{B})) - \mathbf{y}_t)] = \mathbb{E}_t[\mathcal{A}_t^*(\phi(\mathcal{A}_t(\mathbf{B}))) - \mathcal{A}_t^*(\mathbf{y}_t)] \\
&= \mathbb{E}_t[\mathcal{A}_t^*(\phi(\mathcal{A}_t(\mathbf{B}))) - \mathcal{A}_t^*(\phi(\mathcal{A}_t(\mathbf{B})))] = \mathbf{0}.
\end{aligned}
$$

We note that both $\mathcal{A}_t$ and $\mathcal{A}_t^*$ may be computed in time $\mathcal{O}(NC^2d)$. In most of our computations, we average across all available time steps giving a cost of $\mathcal{O}(TNC^2d)$. We also illustrate averaging instead using smaller random sub-windows of the data in Section 4.3. Analogous to Program (5), the VI associated with (8) likewise admits solutions by PG. To illustrate, consider the recurrence

$$\mathbf{B}_{k+1} = \text{Prox}_{\gamma_k\delta_\lambda}(\gamma_k\Psi(\mathbf{B}_k)), \qquad \mathbf{B}_0 \in \{\mathbf{X} \mid \|\mathbf{X}\|_* \leq \lambda\},$$

with step sizes $\{\gamma_k\}$. Note if $\phi := \mathbf{Id}$, the identity function, then vector field associated with the VI corresponds exactly to the *gradient field* of Program (5). In this case, $\Psi(\mathbf{X}) = \nabla_{\mathbf{X}}[\ell(\mathbf{X})]$ and the PG procedures for VI and LS are the same.

**First order methods for monotone VI** To concretely solve the monotone VI outlined in (8), we detail an extragradient scheme with backtracking for nuclear norm constrained VI in Algorithm 1 of Appendix A, which addresses the following general problem

$$\langle \widehat{\Psi}(\mathbf{B}), \mathbf{B} - \mathbf{B}^* \rangle \geq 0 \qquad \forall \mathbf{B} \in \mathcal{X} := \{\mathbf{B} \mid \|\mathbf{B}\|_* \leq \lambda\}$$

where $\widehat{\Psi}$ is an (unbiased estimator of a) $\kappa$-lipschitz continuous monotone vector field $\Psi$ as detailed in Iusem et al. (2019), and which addresses the difficulty that $\kappa$ is in most cases is unknown to us beforehand. The convergence results for this class of algorithm are typical and presented in (Iusem et al., 2019; Gorbunov et al., 2022; Korpelevich, 1976). Namely, for $\epsilon$ error as measured by $\epsilon_{\Psi,\mathcal{X}}(\tilde{\mathbf{B}}_t) = \|\tilde{\mathbf{B}}_t - \text{Prox}_{\mathcal{X}}(\tilde{\mathbf{B}}_t - \Psi(\tilde{\mathbf{B}}_t))\|$ requires iteration complexity on order of $\tilde{\mathcal{O}}(1/\epsilon)$ outer iterations of the extragradient Algorithm 1. The convergence of the algorithm as applied to the vector fields given in (8) in particular may be established similarly to Juditsky & Nemirovski (2019), and when the data follow the true model (1), parameter recovery guarantees can be established similarly to Juditsky et al. (2020). In addition to the ordinary projection, proximal (Nesterov & Nemirovski, 2013; Chen et al., 2017; Nemirovski, 2004; Juditsky et al., 2011), and extragradient schemes, the program induced by (8) can be solved using *projection-free* methods like such conditional gradient (Frank-Wolfe) scheme, which requires only computing singular value/vector per iteration but converges at a slower rate as compared to projection/proximal based schemes and is less stable with respect to stochastic gradients (Hammond, 1985).

**Parameter recovery for symbolic sequences.** As a special case of (8), consider now that each channel represents the probability of emitting a token from syllabary $\{s_c\}_{c=1}^C$ of size $C$. Then each $\mathbf{x}_{i,t}$ represents a probability vector $\sum_c^C x_{i,t,c} = 1$ where $\mathbb{E}[x_{i,t,c}|\mathcal{H}_{i,t}] = \mathbb{P}[x_{i,t,c} = s_c]$. We take the softmax activation function $\sigma(\mathbf{y}) = \text{vec}([\|\exp(\mathbf{y}^{(i)})\|_1^{-1}\exp(\mathbf{y}^{(i)})]_{i=1}^N)$, where $\mathbf{y}^{(i)}$ corresponds to values from the $i^{\text{th}}$ sequence. This problem corresponds to learning representations for different sequences. We illustrate in Section 4.3 the above as applied to learning dynamics for genomics data and natural language, for which autoregressive models have become increasingly popular.

## 4 EXPERIMENTS

We first illustrate parameter recovery using synthetic univariate time-series in Section 4.1. We investigate the choice of nuclear penalty $\lambda$ and the rank of the recovered parameter matrix as it relates to reconstruction quality. Section 4.2 describes benchmarks using real-world time-series data from the UCR Time Series Classification Archive (Dau et al., 2018). We report classification and runtime performance against a number of baselines. Section 4.3 provides two illustrations on embedding of real-world sequence data. We first consider a language representation task where we embed without supervision a series of excerpts taken either from the works of Lewis Carroll or abstracts scraped from arXiv (Carroll, 1865; 1871; Kaggle Team, 2020). In the second illustration, we apply our method to the clustering of gene sequences for strains of *Influenza A* and *Dengue* viruses (Sayers et al., 2022). Appendix A provides implementations for solving the programs in Sections 3.1 and 3.2 in pseudocode and the Julia programming language. The full experimental setup and results are detailed in Appendix B.

In Appendix A we discuss and provide implementations to solve the programs described in Sections 3.1 and 3.2 and describe the experimental setup and results in detail in Appendix B.

### 4.1 PARAMETER RECOVERY WITH SYNTHETIC AUTOREGRESSIVE SEQUENCES

To illustrate parameter recovery across autoregressive sequences, we synthetically generated a set of ten baseline parameters for linear autoregressive sequences of order $d = 15$. Within each class, we then created $N = 300$ sequences of each type, perturbing the baseline coefficients by adding a small amount of noise according to a fixed rule for the set of parameters. We then generated $T = 250$ observations for each sequence according to the autoregressive model in (1) with linear link function $\phi(\mathbf{x}) = \mathbf{x}$. We formed all 120 combinations of $k = 3$ type of sequences from the ten classes and recovered the underlying parameter matrix by solving Program (5) to optimality. We report the data-generating procedure and experimental details in Appendix B.1. To further illustrate parameter recovery with a nonlinear link function, we provide an additional illustration using synthetic symbolic sequences in Appendix B.2

Table 1 reports averages and standard deviations for the relative reconstruction error $\|\mathbf{B}-\widehat{\mathbf{B}}\|_F/\|\mathbf{B}\|_F$, the least squares error of the objective function given in (3), the Adjusted Rand Index (ARI) using k-means clustering with $k = 3$ clusters in the embedding space across the runs, and the number of large singular values (singular values within $10^{-2}$ of the principal singular value). We first learn representations for sequences without nuclear norm constraint.

To illustrate the low-rank matrix recovery, we search across the values of nuclear regularization $\lambda$ via Brent search (Brent, 2002) and report the performance for a close to the optimal value of $\lambda$ with respect to the reconstruction error. When the underlying dynamics share a common low-rank structure, the nuclear constraint effectively leverages the common information shared across the sequences to recover the true parameters more faithfully using the common domain information with comparable error in the least squares sense. Furthermore, we observe the nuclear regularization procedure driving the singular value spectrum to sparsity, with the number of large singular values being much smaller than in the unconstrained case.

To further illustrate, Figure 1 depicts parameter recovery for a collection of sequences with autoregressive order $d = 15$. In the leftmost pane, we report the relative reconstruction error and the number of large singular values across different values of nuclear constraint $\lambda$. In the central pane, we depict the singular value spectra of the true parameter matrix (which has approximately low-rank structure plus noise) and recovered matrices with differing numbers of large singular values. In the third

Table 1: Time series parameter recovery for synthetic autoregressive time-series.

| $\lambda$ Selection | Relative Err. | | LS Err. | | Cluster ARI | | Rank | |
|---|---|---|---|---|---|---|---|---|
| | Avg. | Std. | Avg. | Std. | Avg. | Std. | Avg. | Std. |
| Unconstrained | 0.341 | (0.016) | 79.333 | (0.169) | 0.967 | (0.049) | 14.442 | (1.203) |
| $\arg\min_\lambda \|\widehat{\mathbf{B}}_\lambda - \mathbf{B}\|_F$ | 0.158 | (0.020) | 80.709 | (0.233) | 0.997 | (0.008) | 7.392 | (4.255) |

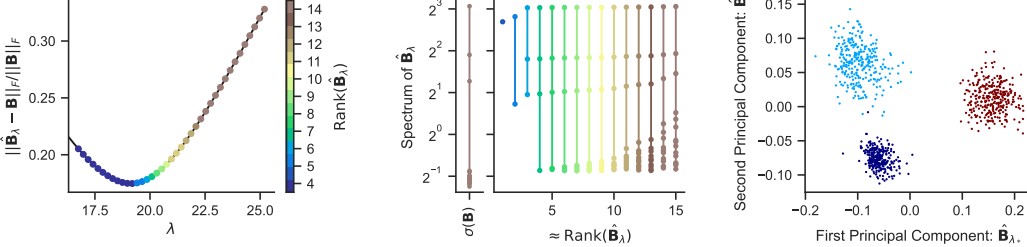

Figure 1: Simulation results: Parameter recovery for a collection of univariate time series drawn from $k = 3$ classes. Left: Relative reconstruction error and approximate rank of recovered parameter matrices across levels of nuclear constraint $\lambda$. Center: Singular values of the true parameter matrix $\mathbf{B}$ and the singular values of the recovered solutions of varying dimensions. Right: First two principal components of the recovered matrix with the smallest reconstruction error and original class labels.

pane, we show the first two principal components (total explained variance $= 0.868$) of the sequence embedding with the smallest reconstruction error along with the original three generating classes of the data. With the introduction of sufficient nuclear regularization, we drastically reduced the reconstruction error of our recovered solution and observed that the solutions with low reconstruction error were of approximately low rank. In the plots of the singular value spectra and the projection of the learned sequence embeddings, we observe that those rank-constrained recoveries effectively recover those large singular values in the spectrum of the true parameter matrix. By contrast, the parameter recoveries performed without or insufficient nuclear constraint fit the noise component of the data, as evident in the distribution of singular values.

## 4.2 REAL TIME-SERIES CLASSIFICATION

Following (Middlehurst et al., 2024; Ma et al., 2019; Yue et al., 2022; Zakaria et al., 2012), we conduct experiments on 36 UCR time series datasets (Dau et al., 2018). Dataset statistics are provided in Appendix B.3.1, with each dataset using its default train/test split. Each time series is re-encoded as a multichannel signal comprising the original signal and its first finite difference. We embed the data without supervision by solving (8) using the extragradient scheme given in Algorithm 1 of Appendix A with a look-back length of $d = 20$, running the algorithm for 256 steps using a linear link function. The value of $\lambda$ is selected via a two-step process: first, bisection identifies when the solution becomes rank-one, and then a grid search refines the choice for rank-constrained parameters. We report results for the $\lambda$ value with the best training performance. Evaluation metrics include ARI (Hubert & Arabie, 1985), Normalized Mutual Information (NMI) (Vinh et al., 2009), macro-F1 score (Fawcett, 2006), accuracy on the test set, and average runtime, including the full grid search and SVD at each step. Runtime improvements using partial SVD computed up to required threshold $\lambda$ are discussed in Appendix B.8.

We compare our method with five representative time series embedding and classification methods: K-nearest neighbors (KNN) using Euclidean ($\ell_2$) distance (Cover & Hart, 1967), KNN with Dynamic Time Warping (DTW) as the distance metric (Müller, 2007), shapeDTW (another method based on DTW but with additional features) (Zakaria et al., 2012; Ye & Keogh, 2009), a dictionary-based method MultiROCKET+Hydra (Dempster et al., 2023), one deep representation method based on contrastive learning (TS2Vec) (Yue et al., 2022) and one based on masked modeling (Ti-MAE) (Cheng et al., 2023). In line with Yue et al. (2022); Franceschi et al. (2019), to evaluate the classification

Table 2: Time series classification performance on UCR time series data of our method vs a number of baselines (higher is better, except for runtime). We outperform simple approaches and perform close to classification using the embeddings from the neural network based TS2Vec but use only 37% of the runtime. The best performing method, MR-Hydra, is a ensemble based on handpicked features tuned specially for time series classification and does not produce latent embeddings.

| Method | ARI | | NMI | | F1 | | Accuracy | | Runtime (Sec) | |
|---|---|---|---|---|---|---|---|---|---|---|
| | Avg. | Std. | Avg. | Std. | Avg. | Std. | Avg. | Std. | Avg. | Std. |
| $\ell_2$+KNN | 0.422 | (0.294) | 0.416 | (0.290) | 0.752 | (0.144) | 0.725 | (0.166) | 0.128 | (0.413) |
| DTW+KNN | 0.447 | (0.303) | 0.435 | (0.300) | 0.766 | (0.150) | 0.738 | (0.170) | 42.988 | (134.320) |
| shapeDTW | 0.470 | (0.307) | 0.460 | (0.299) | 0.773 | (0.152) | 0.746 | (0.178) | 21.871 | (71.655) |
| TiMAE | 0.461 | (0.275) | 0.457 | (0.274) | 0.763 | (0.135) | 0.723 | (0.170) | 1275.712 | (1161.375) |
| TS2Vec | 0.606 | (0.282) | 0.580 | (0.287) | 0.840 | (0.138) | 0.814 | (0.178) | 1,085.092 | (1,408.458) |
| Ours | 0.602 | (0.282) | 0.562 | (0.293) | 0.817 | (0.180) | 0.788 | (0.193) | 400.031 | (677.486) |
| MR-Hydra | 0.682 | (0.273) | 0.656 | (0.285) | 0.877 | (0.121) | 0.851 | (0.162) | 10.197 | (16.459) |

performance on test set for methods which produce embeddings (TS2Vec, Ti-MAE, and our method), we perform cross-validated grid search across KNNs with $k = \{2^i \mid i \in [0, 4]\}$ neighbors or SVMs with RBF kernels with penalty values $c \in \{2^i \mid i \in [-10, 15]\} \cup \infty$. We defer all further details of our experimental setup to Appendix B. To further compare the quality of representations, we provide in Appendix B.5 projections of the learned latent space for various UCR datasets. Table 2 displays the mean and standard deviation across the metrics across the datasets. We provide the detailed results per dataset in Tables 5 and 6 of Appendix B.3. We observe superior performance to baseline methods based on distance metrics, such as Euclidean distance or DTW, and observe performance between that of TiMAE and TS2Vec. We note that for this class of sequence, the heuristic dictionary-based ensemble (MR-HYDRA) outperforms both our approach and the deep-learning-based approaches. However, this method has been tuned specifically for this type of classification problem. By contrast, similar to Yue et al. (2022); Cheng et al. (2023), we consider classification only as one potential downstream task.

## 4.3 SYMBOLIC SEQUENCES: LANGUAGE AND GENOMICS

**Symbolic sequences and language: arXiv abstracts or "Alice in Wonderland"?** To illustrate the capability of our method to learn meaningful representations for sequences with nonlinear dynamics, we first consider an autoregressive language modeling task, drawing textual sequences from three sources: two works by the same author Lewis Caroll — *Alice's Adventures in Wonderland* ($n = 228$) (Carroll, 1865) and *Through the Looking Glass* ($n = 316$) (Carroll, 1871) — and machine learning related abstracts scraped from arXiv ($n = 600$) (Kaggle Team, 2020) (details in Appendix B.6). We embed the sequences without supervision with a lookback of $d = 75$, and in order to reduce the number of symbols in our alphabet and avoid the blowup in the number of channels, we converted each of the sequences into a $c = 4$ symbol code via Huffman coding, based on the overall frequencies of letters in the English language (Cover & Thomas, 2005). We then solve Program induced by (8) using the multichannel measurement operator given in (4) to optimality and using the softmax activation discussed in Section 3.2. We show in Figure 2a the space learned when $\lambda$ was chosen to be sufficiently small as to give a rank three representation of the data. We then project the learned representation via Uniform Manifold Approximation and Projection (UMAP) (McInnes et al., 2018). Two distinct clusters form corresponding to the two different genres of writing, however, whereas the paper abstracts are clearly separable from the works of Lewis Caroll, the two books written by him are not as clearly disambiguable as they are from the same author.

**Virus strain identification from genome sequences** For the final illustration, in line with Millan Arias et al. (2022), the problem of classifying genetic sequences, which allows for the placing of species/variants in the evolutionary context of others (phylogenetic classification). We consider gene sequence data from segment six of the *Influenza A virus* genome ($n = 949$, average sequence length $= 1409$) (Bao et al., 2008) and the complete genome the *Dengue virus* ($n = 1633$, average sequence length $= 10559$) (Hatcher et al., 2017). We consider gene sequences from five strains of Influenza and the four strains of Dengue. Likewise, we provide a detailed overview of the data and learning procedure in Appendix B.7. We encode the genomes in a similar manner

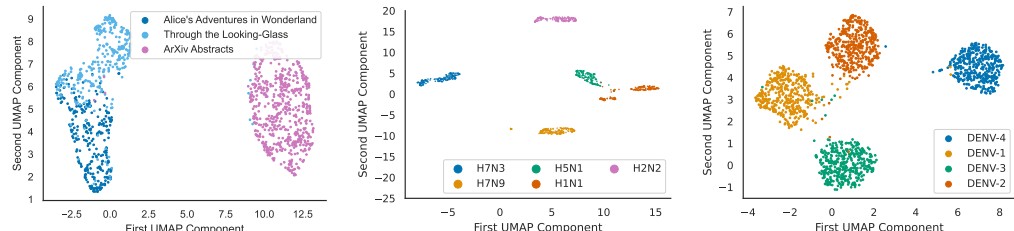

(a) Lewis Caroll or ArXiv abstracts? Genres form clusters. (b) Clustering strains of *Influenza A virus* genome data (segment 6) (c) Clustering using full genome for the four strains of *Dengue Virus*

Figure 2: Learned embeddings for symbolic sequences collections using our method— visualized by UMAP projections shows clear groupings based on sequences with similar underlying dynamics.

as for the natural language illustration, assigning one channel to each nucleotide (A, C, T, G), and encode the presence/absence of each nucleotide at each position via one-hot encoding.

To recover the embedding, we adopt the same softmax activation scheme as described in Section 3.2. Since the genomes are of variable length, we consider a stochastic approximation to the monotone field $\Psi$ (8) by taking the sample average of randomly selected length $G = 800$ sub-windows from each of sequences at each training step. We consider clustering the Influenza and Dengue genome segments individually and report UMAP projections of the learned representations in Figure 2b and 2c, respectively. The dimensions of the learned embeddings are 7 and 20, respectively. In these subspaces, we note the clear grouping of viral strains obtained via solving the stochastic approximation to the VI in (8).

## 5 DISCUSSION

We propose a method to learn embeddings for sequences and time series by framing it as a low-rank matrix recovery problem cast into a VI form. This approach is particularly amenable to settings with partial or limited observations, allowing similar sequences to inform the representation of a focal sequence. Each sequence is modeled with its own autoregressive process through a monotone link function. Our observation model is both a strength and limitation: on one hand is *as general as possible* while still maintaining convexity, and thus flexible enough to handle a number of diverse scenarios — notably probabilistic modeling of symbolic data — and demonstrates empirical performance comparable to methods based on contrastive learning and masked modeling. On the other hand, reliance on convexity to ensure regularity and identifiability limits its ability to capture highly non-convex structures and provide universal approximation guarantees. Our method performs well under low-rank and monotonicity assumptions, is sample-efficient, and is faster in limited-data settings, as shown in most cases. However, its performance declines when these assumptions are violated, seen in certain UCR datasets, where it may be outperformed by energy-based approaches in data-rich scenarios. Future work could explore alternative objectives within the VI framework and non-convex extensions to address these limitations.

### ACKNOWLEDGMENTS

This work is partially supported by NSF DMS-2134037.

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

## A  IMPLEMENTATION DETAILS

### A.1  FIRST ORDER METHODS FOR MONOTONE VI WITH NUCLEAR BALL SETUP

In Algorithm 1, we present a concrete extragradient method with backtracking for nuclear norm constrained VI based primarily on the Algorithms given in (Iusem et al., 2019) and (Duchi et al., 2008).

The monotonicity of $\Psi$ itself when the link function $\phi$ is monotone is readily established by two facts about the calculus of monotone vector fields:

**Affine substitution of argument**  If $\phi(\cdot)$ is monotone vector field on $\mathbb{R}^m$ and $\mathbf{A} \in \mathbb{R}^{n \times m}$ is a matrix, the vector field

$$g(x) = \mathbf{A}\phi(\mathbf{A}^*x + a)$$

is monotone on $\mathbb{R}^n$

**Summation** If $S$ is a Polish space, $\phi(x,s) : \mathbb{R}^m \times S \to \mathbb{R}^m$ is a Borel vector-valued function which is monotone in $x$ for every $s \in S$ and $\mu(ds)$ is a Borel probability measure on $S$ such that the vector field

$$F(x) \equiv \int_S \phi(x,s)\mu(ds)$$

is well defined for all $x$, then $F(\cdot)$ is monotone.

for a more detailed discussion, see Juditsky & Nemirovski (2016). Since the vector field given by the link function $f(\cdot)$ is continuous and monotone, and the expectation is well-defined, the vector field

$$\Psi(\widehat{\mathbf{B}}) = \mathbb{E}_t[\mathcal{A}_t^*(\phi(\mathcal{A}_t(\widehat{\mathbf{B}})) - \mathbf{y}_t)]$$

is monotone and well defined since $\mathcal{A}_t^*(\phi(\mathcal{A}_t(\widehat{\mathbf{B}})) - \mathbf{y_t})$ is monotone for all linear operators $\mathcal{A}_t$ and vectors $\mathbf{y}_t$ by affine substitution, and since expectation can be expressed by definition as an integration (summation in the empirical approximation) with respect to a Borel measure.

---

**Algorithm 1** Extragradient Method with Backtracking for Nuclear Norm constrained VI

---

1: **procedure** BACKTRACKINGEXTRAGRADIENT($\Psi$, $N$, $\lambda$, $\kappa_0$, $\theta$, $\nu$) $\triangleright$ $\Psi$ (stochastic Estimate to) monotone VI, $N > 0$ number of steps, $\lambda > 0$ radius of nuclear ball, $\kappa_0 \in (0,1]$ initial step size, $\theta \in (0,1]$ step size decay parameter, $\nu \in (0, \frac{1}{\sqrt{2}}]$ line search parameter
2:     $\mathbf{B}_1 := \mathbf{0}$
3:     **for** $t := 1, N$ **do**
4:         **for** $j := 1 \dots$ **do**                   $\triangleright$ Line Search
5:             $\kappa := \theta^j \kappa_0$
6:             $\mathbf{R}_t^\kappa := \text{PROXNUC}(\mathbf{B}_t - \kappa(\Psi(\mathbf{B}_t^\kappa)), \lambda)$         $\triangleright$ Extragradient step
7:             **if** $\kappa\|\Psi(\mathbf{R}_t^\kappa) - \Psi(\mathbf{B}_t)\|_F \leq \nu\|\mathbf{R}_t^\kappa - \mathbf{B}_t\|_F$ **then**
8:                 **break**
9:             **end if**
10:         **end for**
11:         $\mathbf{B}_{t+1} := \text{PROXNUC}(\mathbf{B}_t - \kappa\Psi(\mathbf{R}_t^\kappa)\lambda)$
12:     **end for**
13:     **return** $\mathbf{B}_N$
14: **end procedure**
15: **procedure** PROXNUC($\mathbf{A}$,$\lambda$)                 $\triangleright$ Computes $\text{Prox}_{\delta_\lambda}(\mathbf{A})$, $\mathbf{A} \in \mathbb{R}^{m \times n}$
16:     $r := \min(m, n)$
17:     $\mathbf{U} := []; \mathbf{V} := []; \sigma := []$
18:     $cs := 0$                         $\triangleright$ Cumulative sum of singular values
19:     **for** $j := 1, \dots, r$ **do**
20:         Compute $\mathbf{u}_j, \sigma_j, \mathbf{v}_j$               $\triangleright$ From Lanczos iteration on $\mathbf{A}$
21:         **if** $\sigma_j * j \leq (cs + \sigma_j) - \lambda$ **then**
22:             **break**
23:         **end if**
24:         $cs := cs + \sigma_j$
25:         $\mathbf{V} := \begin{bmatrix} \mathbf{V} & \mathbf{v}_j \end{bmatrix}, \; \boldsymbol{\sigma} := \begin{bmatrix} \boldsymbol{\sigma} & \sigma_j \end{bmatrix}, \; \mathbf{U} := \begin{bmatrix} \mathbf{U} & \mathbf{u}_j \end{bmatrix}$
26:     **end for**
27:     $\theta := (cs - \lambda)/j$
28:     **if** $j < r$ **then**
29:         **return** $\mathbf{U} \, \text{diag}(\boldsymbol{\sigma} - \theta)\mathbf{V}^*$
30:     **else**
31:         **return** $\mathbf{A}$
32:     **end if**
33: **end procedure**

---

### A.2 IMPLEMENTATION

We implement Algorithm 1, and associated subroutines (evaluation of the monotone field $\Psi$, as defined in Equation (8), incremental simplex/nuclear ball projection), using the Julia programming language. The implementation is available at `https://github.com/XSpace2013/LowRankTimeSeriesRecovery`.

Table 3: Five classes of sequence generating procedure

| Baseline Coefficients | Perturbation Pattern |
| --- | --- |
| Exponentially Time Decaying | Gaussian |
| Exponentially Time Decaying | $d/3$ Most Recent |
| Exponentially Time Decaying | Uniform $\times$ Fixed Vector |
| Uniform | Gaussian |
| Uniform | Uniform $\times$ Fixed Vector |

## B  DETAILED EXPERIMENTAL SETUP AND RESULTS

We evaluated all experiments and illustrations using a cluster with 24 core Intel Xeon Gold 6226 CPU (2.7 GHZ) processors, and NVIDIA Tesla V100 Graphics coprocessors (16 GB VRAM), and 384 GB of RAM. However, it is also possible to reproduce the results on a standard personal computer.

### B.1  SYNTHETIC TIME SERIES

#### B.1.1  DATA GENERATION

For the synthetic sequence recovery experiment, we adopt the following data-generating procedure: We take the order of the sequences to be $d = 15$, and we generate data according to the following procedure within each of the five generated classes of observations

1. Pick a baseline set of coefficients according to a given random distribution
2. For each of the $N = 300$ sequences to generate, perturb the coefficients according to the pre-specified rule
3. Generate the data matrix of size consisting of $T = 250$ of the $N = 300$ sequences according to the perturbed coefficients such that the data obeys (1) with linear link function $\phi(\mathbf{x}) = \mathbf{x}$. To do so, we seat the first 15 observations using random noise such that $x_{i,t} \sim \mathcal{N}(\mu = 0, \sigma^2 = 1), \forall t \in [1, d], i \in [N]$. Then each successive entry $x_{i,t}, t \in [d+1, T]$ is then given by taking $x_{i,t} = \sum_{s=1}^{d} b_{i,s} x_{i,t-s} + \epsilon_{i,t}, \epsilon_{i,t} \sim \mathcal{N}(\mu = 0, \sigma^2 = 0.02)$.

We draw the ten generated classes of data from the following five generation procedures given in Table 3. We use each procedure twice to generate the ten classes of data. We denote the coefficients common to the sequences (for some class) as $\mathbf{b}_{\text{common}}$, and the coefficients for the $i^{\text{th}}$ sequence in said class as $\mathbf{b}_i$.

The baseline coefficients generation methods are given as:

**Exponentially Time Decaying:** $b_{\text{common},s} = Z\gamma^s/(\sum_{j=1}^{d} \gamma^j) \qquad Z \sim \text{Uniform}([0, 1]), \forall s \in [d]$

**Uniform:** $b_{\text{common},s} = Z \qquad Z \sim \text{Uniform}([0, 1/2d]), \forall s \in [d]$

and the perturbation methods are given as:

**Gaussian:** $\mathbf{b}_i = \mathbf{b}_{\text{common}} + \mathbf{Z} \qquad Z_j \sim \mathcal{N}(\mu = 0, \sigma^2 = 0.02)$

$d/3$ **Most Recent:** $\mathbf{b}_i = \mathbf{b}_{\text{common}} + \mathbf{Z} \qquad Z_j \sim \begin{cases} \mathcal{N}(\mu = 0, \sigma^2 = 0.02) & j < \lceil d/3 \rceil \\ 0 \end{cases}, \forall i \in [N]$

**Uniform $\times$ Fixed Vector:** $\mathbf{b}_i = \mathbf{b}_{\text{common}} + \theta\mathbf{v} \qquad \theta \sim \text{Uniform}([-1, 1]), \|\mathbf{v}\| = 1, \forall i \in [N]$ \$$\mathbf{v}$ chosen uniformly on a unit hypersphere, and is the same for all sequences generated in the class)

#### B.1.2  PARAMETER RECOVERY

For the parameter recovery experiment, we take all $\binom{10}{3} = 120$ combinations of $k = 3$ sequences from the 10 classes and concatenate the generated sequences to form a matrix of 900 observations.

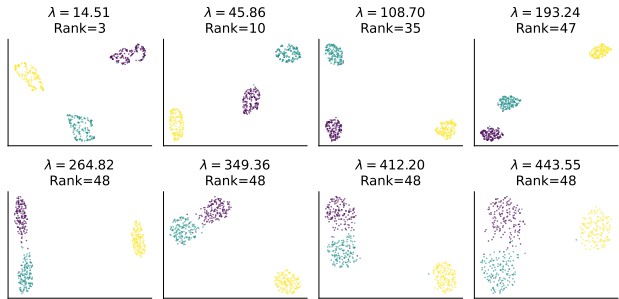

Figure 3: Learned representations for symbolic sequences generated from Hidden Markov Models (HMMs), recovered by solving Program (5) with varying nuclear constraints $\lambda$, visualized using UMAP projection.

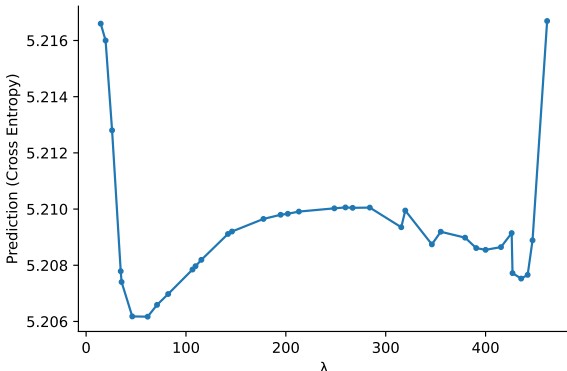

Figure 4: Cross entropy as function of $\lambda$ across for symbolic sequence generated by HMMs.

We then recover the baseline coefficient matrix $\mathbf{B} \in \mathbb{R}^{15 \times 900}$. To recover the parameters for each sequence, we solve the program given in (5) to optimality for differing levels of $\lambda$ (using a standard convex solver by contrast to Algorithm 1). To find which levels of $\lambda$ to solve for, we first solve the unconstrained version of the problem. We then compute the nuclear norm of recovered $\|\widehat{\mathbf{B}}\|_*$. We then successively search for the optimal $\lambda^*$ on the interval $[0, \|\widehat{\mathbf{B}}\|_*]$ using the relative reconstruction error $\|\mathbf{B} - \widehat{\mathbf{B}}\|_F / \|\mathbf{B}\|_F$ as the objective until we achieve an absolute tolerance of $10^{-3}$. We report the results in Table 1 using the matrices $\widehat{\mathbf{B}}$ and $\widehat{\mathbf{B}}_{\lambda^*}$ across the 120 runs. For Figure 1, we use data drawn from the classes uniform baseline coefficients with Gaussian perturbation, exponentially time-decaying baseline coefficients with Gaussian perturbation; and exponentially time-decaying baseline coefficients with uniform*fixed vector perturbation. We find the $\lambda^*$ via Brent search, and we sweep 40 values of $\lambda \in [16.3, 25.2]$ (the right bound corresponding to the value of $\|\widehat{\mathbf{B}}\|_*$) to produce Figure 1. When reporting the spectra of singular values, we pick solutions that correspond to the largest value of $\lambda$ we have for some fixed number of large singular values. Finally, we depict the principal components according to the recovered matrix $\widehat{\mathbf{B}}_{\lambda^*}$, which is optimal in the sense of the reconstruction error (found via Brent search).

## B.2 SYNTHETIC SEQUENCE DATA

### B.2.1 DATA GENERATION

To illustrate representation learning and parameter recovery in the nonlinear case, we simulate an HMMs with four hidden and four observed states. Transitions between states are random, and the emission matrix assigns a $0.9$ probability to emitting the symbol corresponding to the hidden state,

Table 4: Basic statistics for the UCR time-series classification benchmark data

|  | Training Samples | Testing Samples | Length | Classes |
|---|---|---|---|---|
| ArrowHead | 36 | 175 | 251 | 3 |
| Beef | 30 | 30 | 470 | 5 |
| BeetleFly | 20 | 20 | 512 | 2 |
| BirdChicken | 20 | 20 | 512 | 2 |
| Car | 60 | 60 | 577 | 4 |
| ChlorineConc. | 467 | 3840 | 166 | 3 |
| Coffee | 28 | 28 | 286 | 2 |
| DiatomsizeReduction | 16 | 306 | 345 | 4 |
| Dist.Pha.Outln.AgeGrp. | 400 | 139 | 80 | 3 |
| Dist.Pha.Outln.Correct | 600 | 276 | 80 | 2 |
| ECG200 | 100 | 100 | 96 | 2 |
| ECGFiveDays | 23 | 861 | 136 | 2 |
| GunPoint | 50 | 150 | 150 | 2 |
| Ham | 109 | 105 | 431 | 2 |
| Herring | 64 | 64 | 512 | 2 |
| Lightning2 | 60 | 61 | 637 | 2 |
| Meat | 60 | 60 | 448 | 3 |
| Mid.Pha.Outln.AgeGrp. | 400 | 154 | 80 | 3 |
| Mid.Pha.Outln.Correct | 600 | 291 | 80 | 2 |
| Mid.PhalanxTW | 399 | 154 | 80 | 6 |
| MoteStrain | 20 | 1252 | 84 | 2 |
| OSULeaf | 200 | 242 | 427 | 6 |
| Plane | 105 | 105 | 144 | 7 |
| Prox.Pha.Outln.AgeGrp. | 400 | 205 | 80 | 3 |
| Prox.PhalanxTW | 400 | 205 | 80 | 6 |
| SonyAIBORobotSurf.1 | 20 | 601 | 70 | 2 |
| SonyAIBORobotSurf.2 | 27 | 953 | 65 | 2 |
| SwedishLeaf | 500 | 625 | 128 | 15 |
| Symbols | 25 | 995 | 398 | 6 |
| ToeSegmentation1 | 40 | 228 | 277 | 2 |
| ToeSegmentation2 | 36 | 130 | 343 | 2 |
| TwoPatterns | 1000 | 4000 | 128 | 4 |
| TwoLeadECG | 23 | 1139 | 82 | 2 |
| Wafer | 1000 | 6164 | 152 | 2 |
| Wine | 57 | 54 | 234 | 2 |
| WordSynonyms | 267 | 638 | 270 | 25 |

with equal probability among other symbols. The models are initialized randomly and simulated for 100 time steps, generating 150 trajectories for each random HMM.

### B.2.2 PARAMETER RECOVERY WITH SEQUENCE DATA

For parameter recovery demonstrating a nonlinear sequence dynamics, we sample three random HMMs and aggregate their realizations (450 sequences in total). We solve Program induced by (8) using these realizations, sweeping across different values of $\lambda$.

Figure 3 depicts the learned low-rank space projected into two dimensions using UMAP, showing the choice of $\lambda$ and the approximate rank of the space. Figure 4 presents the relationship between the nuclear ball size ($\lambda$), and the loss as measured cross entropy of the trained parametric model $\phi(\mathbf{R}_i \boldsymbol{\xi}_{i,t})$ with the data across the entire observation timeframe.

The results largely align with the discussion in Section 4.1. Small amounts of nuclear regularization improve the quality of sequence representations and improve the prediction performance, as the information from other sequences becomes incorporated to aid in prediction. The sequences sequences also cluster within a constrained representational subspace. The data remains well-separated even in low-dimensional spaces (e.g., rank 3).

### B.3 UCR TIME SERIES

### B.3.1 DATA OVERVIEW

We compare our method with the following representative time series clustering methods. In line with Ma et al. (2019) we selected 36 of the univariate UCR time series classification datasets [1]. We report basic statistics (training samples, testing samples, length, and number of classes) of the datasets in Table 4.

---

[1] `https://www.cs.ucr.edu/%7Eeamonn/time_series_data_2018/`

### B.3.2 OVERVIEW OF EVALUATION METHODS

We evaluate the classification performance using the following methods:

**ARI:** Similarity of learned and ground truth assignments (Vinh et al., 2009). For matched clustering partitions

$$\text{RI} = (a + b)/\binom{N}{2}, \qquad \text{ARI} = \frac{\text{RI} - \mathbb{E}[\text{RI}]}{\max(\text{RI}) - \mathbb{E}[RI]]}$$

**NMI:** The mutual information between the true class labels and the cluster assignments, normalized by the entropy of the true labels and the cluster assignments (Vinh et al., 2009).

$$\text{NMI} = \frac{2I(X;Y)}{H(X) + H(Y)}$$

where $X, Y$ are the true and assigned labels, $H(X)$ is the entropy of $X$, and $I(X;Y)$ is the mutual information between $X$ and $Y$.

**Accuracy:** Proportion of correct predictions to a total number of predictions.

**F1:** Harmonic mean of the precision and recall. In the multiclass case, we take the macro average by calculating the metric for each label and computing their unweighted mean.

$$F_1 = \frac{2 \times \text{TP}}{2 \times \text{TP} + \text{FP} + \text{FN}}$$

**Runtime:** Runtime of the algorithm in terms the user CPU time in the computational setting described in 4. If the method is GPU accelerated we report the user CPU/GPU time spent in the routine.

### B.3.3 OVERVIEW OF METHODS

In addition to our method, we evaluate the performance of the following baseline methods

$\ell_2$**+KNN** K-Nearest Neighbors Classification with the distance metric as the Euclidean distance between two time-series treating the entire observation sequence as a high dimensional vector (Cover & Hart, 1967).

**DTW+KNN** K-Nearest Neighbors Classification with the distance metric calculated according to DTW (Müller, 2007), which aims to align the two given sequences by solving the following program

$$\text{DTW}_q(\mathbf{x}, \mathbf{x}') = \min_{\pi \in \mathcal{A}(\mathbf{x}, \mathbf{x}')} \langle A_\pi, D_q(\mathbf{x}, \mathbf{x}') \rangle^{1/q}.$$

The set $\mathcal{A}(\mathbf{x}, \mathbf{x}')$ is the set of all admissible paths as represented by boolean matrices. Non-zero entries correspond to matching time series elements in the path. A path is admissible if the beginning and end of the time series are matched together, the sequence is monotone in both $i$ and $j$, and all entries appear at least once. We take $q = 2$ as the Euclidean metric.

**shapeDTW** Extension to DTW scheme by incorporating point-wise local structures into the matching procedure (Zhao & Itti, 2018). Examples of such *shape descriptors* include data itself, a rolling average of, a discrete wavelet transform, and a finite difference/derivative. Finally, the encoded sequences are then aligned by DTW and used for nearest neighbor classification.

**MR-Hydra** Combination of dictionary-based Multirocket and Hydra algorithms for time series classification, extracts and counts symbolic patterns using competing convolutional kernels (Dempster et al., 2023; Tan et al., 2022).

**TS2Vec** Construct an encoder network for time series embedding based on hierarchical contrastive learning (Yue et al., 2022). The discrimination is done both between sequences and within the sequences themselves. The encoder network consists of an input projection layer, a timestamp masking module, and a dilated convolutional module, and is optimized jointly with temporal and cross-sequence contrastive loss.

**Ti-MAE** Like all auto-encoding models, an encoder network maps a time series signal into a latent representational space, and then a decoder aims to reconstruct the original sequence from the representational space. Once the input has been tokenized, a random sample of tokens are masked, and then the decoder attempts to reconstruct the time series optimizing the on self supervised reconstruction loss (Cheng et al., 2023).

### B.3.4 DETAILED EXPERIMENTAL PROCEDURE

We split our data into testing and training splits according to those given by the UCR repository. For the methods that directly perform classification (KNN, shapeDTW, Inception Time), we train on the test set and then report the performance on the training set. In line with Yue et al. (2022); Franceschi et al. (2019), to evaluate the classification performance on test set for methods which produce embeddings (TS2Vec, Ti-MAE and our method), we perform cross-validated grid search (based on $k = 5$ folds) across KNNs with $k = \{2^i \mid i \in [0, 4]\}$ neighbors or SVMs with RBF kernels with penalty values $c \in \{2^i \mid i \in [-10, 15]\} \cup \infty$. For the KNN-based methods, we do the same grid search as outlined above. For our own method, we also grid search across parameters of $\lambda$ and report the performance for the best choice under rank constraint. To find the embedding, we run Algorithm 1 for 256 iterations.

### B.4 DETAILED NUMERICAL RESULTS

In Tables 5 and 6, we present the classification performance for the discussed metrics for the evaluated methods for each of the tested UCR datasets.

### B.5 REPRESENTATION VISUALIZATION

In Figure 5 we provide a comparison of the embedding quality for some of the recovered real-time series for our method (left) as compared two recent and popular neural-network based time-series representation learning methods — TS2Vec (Yue et al., 2022) (center) and Ti-MAE (Cheng et al., 2023) (right). We plot the category of the data in color, though this information is not provided to the models during training. We note that our method produces similar quality embeddings to these approaches, with better separation of the data according to category in the low sample cases (e.g., BeetleFly, BirdChicken). For all three models, there exist cases, where the separation is worse for certain datasets, for example Ti-MAE on TwoLeadECG, our method on TwoPatterns.

### B.6 NATURAL LANGUAGE EMBEDDING

To acquire the data, we retrieved the raw text of *Alice's Adventures in Wonderland* and *Through the Looking Glass* from Project Gutenberg [2]. For the paper abstracts, we used the training portion of the `ML-ArXiv-Papers` dataset [3]. For each dataset, we stripped all non ASCII characters and uncommon punctuation (<,>, `, =, |, ?, &, [, ], *, , !, #, @, and ").

After acquiring the data, we then encoded using a Huffman tree with $n = 4$ symbols derived from the frequency of letters in our corpora. We treated each abstract as a document and considered 500-character chunks of the two books. We rejected abstracts containing less than 500 words. After encoding the sequences using the Huffman code, we cut off each sequence at 1000 coded symbols and rejected all sequences less than this length after coding. This left us with $n = 228$ samples from "Alice's Adventures in Wonderland", $n = 316$ samples from "Through the Looking Glass", and $n = 600$ machine learning-related ArXiv abstracts.

To learn the embedding, we use the method described in Appendix A.1 and grid searched across values of $\lambda$ for 512 steps using the softmax link function described in Section 3.2.

### B.7 GENE SEQUENCE EMBEDDING

**Data acquisition and processing** In line with Millan Arias et al. (2022), we downloaded viral genome sequences for two different kinds of human viruses: *Influenza A virus* and *Dengue virus*. We consider different strains of each virus in addition to the species as a whole. We provide a textual description below. In Table 7, we provide summary statistics, including the number of sequences in the strain, the average and standard deviation of the sequence lengths, and the length of the shortest and longest sequences in the strains.

---

[2]`https://www.gutenberg.org`
[3]`https://www.kaggle.com/datasets/Cornell-University/arxiv`

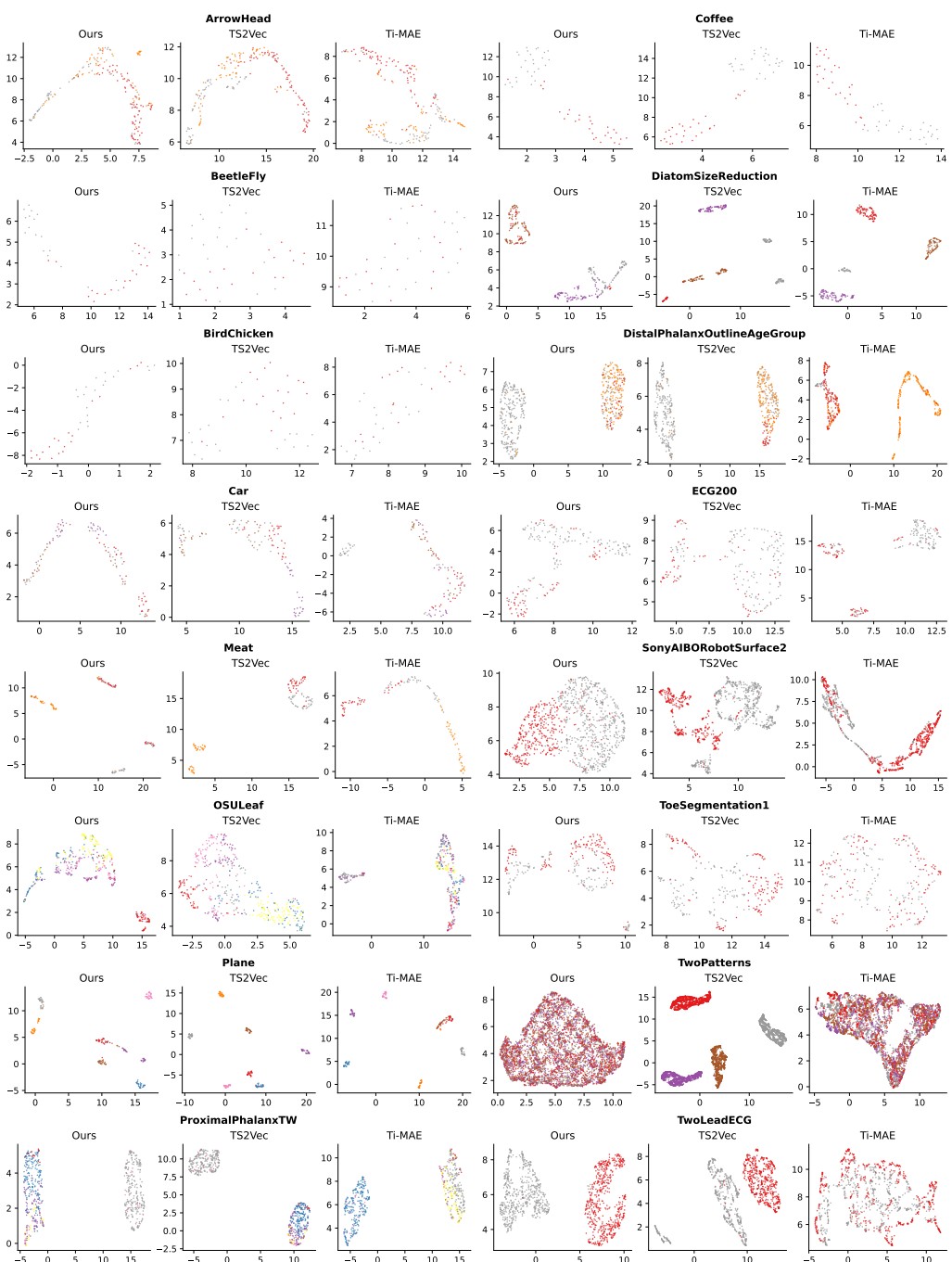

Figure 5: UMAP projections of learned embeddings for UCR datasets by our method, TS2Vec (Yue et al., 2022), and Ti-MAE (Cheng et al., 2023)

Table 5: Detailed results per UCR dataset (Part I)

| Dataset | Method | ARI | NMI | Acc. | F1 | RT | Method | ARI | NMI | Acc. | F1 | RT |
|---|---|---|---|---|---|---|---|---|---|---|---|---|
| ArrowHead | $\ell_2+$ | 0.482 | 0.453 | 0.800 | 0.800 | 0.007 | MR | 0.629 | 0.585 | 0.863 | 0.863 | 3.183 |
| Beef | KNN | 0.322 | 0.518 | 0.667 | 0.672 | 0.001 | Hydra | 0.454 | 0.627 | 0.767 | 0.768 | 1.967 |
| BeetleFly | | 0.219 | 0.344 | 0.750 | 0.733 | 0.001 | | 0.621 | 0.619 | 0.900 | 0.899 | 1.532 |
| BirdChicken | | -0.044 | 0.007 | 0.550 | 0.549 | 0.001 | | 0.621 | 0.619 | 0.900 | 0.899 | 1.536 |
| Car | | 0.403 | 0.477 | 0.733 | 0.737 | 0.003 | | 0.830 | 0.860 | 0.933 | 0.933 | 4.023 |
| ChlorineConc. | | 0.231 | 0.157 | 0.650 | 0.610 | 0.633 | | 0.472 | 0.373 | 0.789 | 0.753 | 57.36 |
| Coffee | | 1.000 | 1.000 | 1.000 | 1.000 | 0.001 | | 1.000 | 1.000 | 1.000 | 1.000 | 1.411 |
| DiatomsizeReduction | | 0.872 | 0.830 | 0.935 | 0.883 | 0.004 | | 0.921 | 0.896 | 0.964 | 0.947 | 6.537 |
| Dist.Pha.Outln.AgeGrp. | | 0.190 | 0.224 | 0.626 | 0.613 | 0.019 | | 0.383 | 0.404 | 0.770 | 0.775 | 4.794 |
| Dist.Pha.Outln.Correct | | 0.181 | 0.137 | 0.717 | 0.684 | 0.054 | | 0.366 | 0.286 | 0.804 | 0.790 | 8.056 |
| ECG200 | | 0.571 | 0.445 | 0.880 | 0.868 | 0.004 | | 0.667 | 0.542 | 0.910 | 0.902 | 1.765 |
| ECGFiveDays | | 0.352 | 0.304 | 0.797 | 0.794 | 0.011 | | 1.000 | 1.000 | 1.000 | 1.000 | 7.914 |
| GunPoint | | 0.681 | 0.578 | 0.913 | 0.913 | 0.004 | | 1.000 | 1.000 | 1.000 | 1.000 | 2.064 |
| Ham | | 0.031 | 0.029 | 0.600 | 0.600 | 0.006 | | 0.229 | 0.177 | 0.743 | 0.742 | 5.055 |
| Herring | | -0.015 | 0.003 | 0.516 | 0.516 | 0.003 | | 0.207 | 0.155 | 0.734 | 0.726 | 3.825 |
| Lightning2 | | 0.246 | 0.193 | 0.754 | 0.750 | 0.003 | | 0.104 | 0.084 | 0.672 | 0.665 | 5.208 |
| Meat | | 0.799 | 0.797 | 0.933 | 0.935 | 0.003 | | 0.810 | 0.808 | 0.933 | 0.933 | 3.096 |
| Mid.Pha.Outln.AgeGrp. | | 0.055 | 0.026 | 0.519 | 0.443 | 0.021 | | 0.092 | 0.071 | 0.591 | 0.491 | 4.599 |
| Mid.Pha.Outln.Correct | | 0.280 | 0.208 | 0.766 | 0.756 | 0.057 | | 0.475 | 0.372 | 0.845 | 0.842 | 8.154 |
| Mid.PhalanxTW | | 0.379 | 0.367 | 0.513 | 0.382 | 0.020 | | 0.383 | 0.433 | 0.513 | 0.339 | 5.382 |
| MoteStrain | | 0.573 | 0.467 | 0.879 | 0.877 | 0.015 | | 0.794 | 0.699 | 0.946 | 0.945 | 7.062 |
| OSULeaf | | 0.298 | 0.383 | 0.521 | 0.525 | 0.023 | | 0.921 | 0.919 | 0.963 | 0.956 | 10.34 |
| Plane | | 0.919 | 0.943 | 0.962 | 0.963 | 0.005 | | 1.000 | 1.000 | 1.000 | 1.000 | 2.505 |
| Prox.Pha.Outln.AgeGrp. | | 0.492 | 0.422 | 0.785 | 0.693 | 0.027 | | 0.662 | 0.564 | 0.868 | 0.797 | 5.489 |
| Prox.PhalanxTW | | 0.584 | 0.566 | 0.707 | 0.444 | 0.027 | | 0.718 | 0.671 | 0.805 | 0.490 | 4.922 |
| SonyAIBORobotSurf.1 | | 0.148 | 0.280 | 0.696 | 0.688 | 0.007 | | 0.598 | 0.570 | 0.887 | 0.887 | 3.396 |
| SonyAIBORobotSurf.2 | | 0.514 | 0.395 | 0.859 | 0.849 | 0.013 | | 0.782 | 0.682 | 0.942 | 0.940 | 4.828 |
| SwedishLeaf | | 0.629 | 0.761 | 0.789 | 0.782 | 0.109 | | 0.950 | 0.965 | 0.976 | 0.977 | 10.78 |
| Symbols | | 0.791 | 0.843 | 0.899 | 0.898 | 0.017 | | 0.955 | 0.954 | 0.981 | 0.981 | 21.89 |
| ToeSegmentation1 | | 0.126 | 0.095 | 0.680 | 0.675 | 0.006 | | 0.832 | 0.782 | 0.956 | 0.956 | 4.760 |
| ToeSegmentation2 | | 0.340 | 0.244 | 0.808 | 0.744 | 0.003 | | 0.640 | 0.464 | 0.915 | 0.866 | 3.608 |
| TwoPatterns | | 0.770 | 0.726 | 0.907 | 0.906 | 1.328 | | 1.000 | 1.000 | 1.000 | 1.000 | 50.14 |
| TwoLeadECG | | 0.244 | 0.217 | 0.747 | 0.741 | 0.015 | | 0.993 | 0.983 | 0.998 | 0.998 | 6.182 |
| Wafer | | 0.971 | 0.923 | 0.995 | 0.988 | 2.088 | | 0.998 | 0.993 | 1.000 | 0.999 | 76.27 |
| Wine | | 0.031 | 0.036 | 0.611 | 0.611 | 0.002 | | 0.720 | 0.687 | 0.926 | 0.926 | 1.718 |
| WordSynonyms | | 0.537 | 0.571 | 0.618 | 0.465 | 0.069 | | 0.725 | 0.753 | 0.777 | 0.658 | 15.76 |
| ArrowHead | DTW+ | 0.312 | 0.282 | 0.703 | 0.700 | 2.139 | TS2 | 0.480 | 0.462 | 0.794 | 0.794 | 81.08 |
| Beef | KNN | 0.276 | 0.490 | 0.633 | 0.629 | 1.158 | Vec | 0.284 | 0.494 | 0.667 | 0.670 | 109.0 |
| BeetleFly | | 0.131 | 0.275 | 0.700 | 0.670 | 0.618 | | 0.800 | 0.761 | 0.950 | 0.950 | 79.59 |
| BirdChicken | | 0.212 | 0.221 | 0.750 | 0.744 | 0.606 | | 0.621 | 0.619 | 0.900 | 0.899 | 80.60 |
| Car | | 0.446 | 0.501 | 0.733 | 0.728 | 7.565 | | 0.709 | 0.787 | 0.867 | 0.867 | 298.9 |
| ChlorineConc. | | 0.231 | 0.154 | 0.648 | 0.607 | 247.313 | | 0.432 | 0.333 | 0.764 | 0.730 | 2439 |
| Coffee | | 1.000 | 1.000 | 1.000 | 1.000 | 0.336 | | 0.857 | 0.811 | 0.964 | 0.964 | 129.1 |
| DiatomsizeReduction | | 0.938 | 0.921 | 0.967 | 0.942 | 3.272 | | 0.968 | 0.952 | 0.984 | 0.973 | 87.44 |
| Dist.Pha.Outln.AgeGrp. | | 0.389 | 0.368 | 0.770 | 0.763 | 1.804 | | 0.272 | 0.277 | 0.705 | 0.699 | 2046 |
| Dist.Pha.Outln.Correct | | 0.183 | 0.132 | 0.717 | 0.690 | 5.382 | | 0.246 | 0.176 | 0.750 | 0.737 | 2882 |
| ECG200 | | 0.280 | 0.192 | 0.770 | 0.749 | 0.468 | | 0.540 | 0.417 | 0.870 | 0.858 | 463.5 |
| ECGFiveDays | | 0.286 | 0.252 | 0.768 | 0.763 | 1.848 | | 0.991 | 0.979 | 0.998 | 0.998 | 80.84 |
| GunPoint | | 0.659 | 0.557 | 0.907 | 0.907 | 0.847 | | 0.973 | 0.949 | 0.993 | 0.993 | 234.7 |
| Ham | | -0.005 | 0.003 | 0.467 | 0.467 | 12.116 | | 0.210 | 0.168 | 0.733 | 0.733 | 526.1 |
| Herring | | -0.012 | 0.001 | 0.531 | 0.520 | 6.536 | | 0.064 | 0.047 | 0.641 | 0.625 | 326.8 |
| Lightning2 | | 0.537 | 0.480 | 0.869 | 0.864 | 8.898 | | 0.318 | 0.252 | 0.787 | 0.783 | 301.2 |
| Meat | | 0.799 | 0.797 | 0.933 | 0.935 | 4.497 | | 0.687 | 0.714 | 0.883 | 0.883 | 300.1 |
| Mid.Pha.Outln.AgeGrp. | | 0.024 | 0.022 | 0.500 | 0.411 | 2.066 | | 0.038 | 0.029 | 0.519 | 0.426 | 2042 |
| Mid.Pha.Outln.Correct | | 0.153 | 0.109 | 0.698 | 0.691 | 5.663 | | 0.385 | 0.297 | 0.811 | 0.808 | 3055 |
| Mid.PhalanxTW | | 0.380 | 0.368 | 0.506 | 0.374 | 1.996 | | 0.420 | 0.404 | 0.545 | 0.396 | 1997 |
| MoteStrain | | 0.448 | 0.351 | 0.835 | 0.834 | 0.900 | | 0.528 | 0.424 | 0.863 | 0.862 | 86.43 |
| OSULeaf | | 0.309 | 0.392 | 0.591 | 0.588 | 50.818 | | 0.644 | 0.671 | 0.822 | 0.799 | 1071 |
| Plane | | 1.000 | 1.000 | 1.000 | 1.000 | 1.172 | | 1.000 | 1.000 | 1.000 | 1.000 | 541.7 |
| Prox.Pha.Outln.AgeGrp. | | 0.504 | 0.430 | 0.805 | 0.716 | 2.696 | | 0.506 | 0.437 | 0.780 | 0.689 | 2051 |
| Prox.PhalanxTW | | 0.644 | 0.587 | 0.756 | 0.511 | 2.673 | | 0.674 | 0.628 | 0.771 | 0.562 | 2052 |
| SonyAIBORobotSurf.1 | | 0.200 | 0.316 | 0.725 | 0.721 | 0.309 | | 0.588 | 0.571 | 0.884 | 0.883 | 84.73 |
| SonyAIBORobotSurf.2 | | 0.435 | 0.324 | 0.831 | 0.817 | 0.571 | | 0.671 | 0.584 | 0.910 | 0.907 | 125.5 |
| SwedishLeaf | | 0.639 | 0.770 | 0.792 | 0.787 | 25.957 | | 0.875 | 0.916 | 0.936 | 0.937 | 2573 |
| Symbols | | 0.891 | 0.913 | 0.950 | 0.949 | 21.459 | | 0.928 | 0.936 | 0.969 | 0.969 | 132.7 |
| ToeSegmentation1 | | 0.293 | 0.260 | 0.772 | 0.762 | 3.704 | | 0.816 | 0.723 | 0.952 | 0.952 | 213.5 |
| ToeSegmentation2 | | 0.398 | 0.249 | 0.838 | 0.764 | 2.904 | | 0.714 | 0.631 | 0.931 | 0.899 | 170.8 |
| TwoPatterns | | 1.000 | 1.000 | 1.000 | 1.000 | 330.318 | | 0.973 | 0.964 | 0.990 | 0.990 | 5216 |
| TwoLeadECG | | 0.654 | 0.564 | 0.904 | 0.904 | 0.901 | | 0.958 | 0.918 | 0.989 | 0.989 | 87.20 |
| Wafer | | 0.867 | 0.748 | 0.980 | 0.944 | 720.459 | | 0.942 | 0.868 | 0.991 | 0.976 | 5386 |
| Wine | | 0.003 | 0.016 | 0.574 | 0.574 | 0.860 | | 0.339 | 0.271 | 0.796 | 0.796 | 303.5 |
| WordSynonyms | | 0.575 | 0.600 | 0.649 | 0.533 | 66.754 | | 0.349 | 0.421 | 0.522 | 0.309 | 1407 |

Table 6: Detailed results per UCR dataset (Part II)

| Dataset | Method | ARI | NMI | Acc. | F1 | RT | Method | ARI | NMI | Acc. | F1 | RT |
|---|---|---|---|---|---|---|---|---|---|---|---|---|---|
| ArrowHead | shape | 0.521 | 0.492 | 0.817 | 0.818 | 0.672 | Ours | 0.336 | 0.306 | 0.720 | 0.720 | 136.7 |
| Beef | DTW | 0.322 | 0.518 | 0.667 | 0.672 | 0.214 | | 0.453 | 0.654 | 0.733 | 0.736 | 70.59 |
| BeetleFly | | 0.219 | 0.344 | 0.750 | 0.733 | 0.107 | | 1.000 | 1.000 | 1.000 | 1.000 | 51.50 |
| BirdChicken | | -0.044 | 0.007 | 0.550 | 0.549 | 0.106 | | 1.000 | 1.000 | 1.000 | 1.000 | 51.92 |
| Car | | 0.560 | 0.585 | 0.817 | 0.815 | 1.077 | | 0.372 | 0.457 | 0.667 | 0.648 | 173.1 |
| ChlorineConc. | | 0.199 | 0.133 | 0.628 | 0.587 | 120.800 | | 0.537 | 0.414 | 0.811 | 0.782 | 2252 |
| Coffee | | 1.000 | 1.000 | 1.000 | 1.000 | 0.105 | | 1.000 | 1.000 | 1.000 | 1.000 | 39.92 |
| DiatomsizeReduction | | 0.921 | 0.890 | 0.958 | 0.921 | 0.849 | | 0.818 | 0.865 | 0.882 | 0.704 | 331.7 |
| Dist.Pha.Outln.AgeGrp. | | 0.209 | 0.251 | 0.633 | 0.615 | 1.584 | | 0.323 | 0.402 | 0.741 | 0.748 | 103.9 |
| Dist.Pha.Outln.Correct | | 0.188 | 0.140 | 0.721 | 0.690 | 4.733 | | 0.316 | 0.233 | 0.783 | 0.772 | 158.0 |
| ECG200 | | 0.541 | 0.420 | 0.870 | 0.860 | 0.397 | | 0.668 | 0.554 | 0.910 | 0.904 | 43.80 |
| ECGFiveDays | | 0.705 | 0.605 | 0.920 | 0.920 | 1.279 | | 0.765 | 0.666 | 0.937 | 0.937 | 306.1 |
| GunPoint | | 0.845 | 0.761 | 0.960 | 0.960 | 0.517 | | 0.845 | 0.761 | 0.960 | 0.960 | 70.03 |
| Ham | | 0.031 | 0.029 | 0.600 | 0.600 | 2.714 | | 0.160 | 0.124 | 0.705 | 0.704 | 265.6 |
| Herring | | -0.012 | 0.006 | 0.531 | 0.531 | 1.129 | | 0.176 | 0.128 | 0.719 | 0.688 | 162.0 |
| Lightning2 | | 0.358 | 0.299 | 0.803 | 0.797 | 1.345 | | 0.399 | 0.320 | 0.820 | 0.817 | 230.8 |
| Meat | | 0.799 | 0.797 | 0.933 | 0.935 | 0.860 | | 0.856 | 0.841 | 0.950 | 0.950 | 133.3 |
| Mid.Pha.Outln.AgeGrp. | | 0.053 | 0.022 | 0.513 | 0.432 | 1.722 | | 0.184 | 0.143 | 0.649 | 0.523 | 106.6 |
| Mid.Pha.Outln.Correct | | 0.281 | 0.207 | 0.766 | 0.759 | 5.110 | | 0.410 | 0.324 | 0.821 | 0.813 | 177.9 |
| Mid.PhalanxTW | | 0.357 | 0.360 | 0.487 | 0.361 | 1.740 | | 0.362 | 0.436 | 0.591 | 0.334 | 107.5 |
| MoteStrain | | 0.573 | 0.467 | 0.879 | 0.877 | 0.804 | | 0.212 | 0.175 | 0.731 | 0.731 | 243.3 |
| OSULeaf | | 0.316 | 0.411 | 0.566 | 0.567 | 10.754 | | 0.600 | 0.625 | 0.810 | 0.798 | 584.7 |
| Plane | | 0.937 | 0.961 | 0.971 | 0.972 | 0.644 | | 1.000 | 1.000 | 1.000 | 1.000 | 74.50 |
| Prox.Pha.Outln.AgeGrp. | | 0.482 | 0.399 | 0.780 | 0.688 | 2.244 | | 0.681 | 0.584 | 0.883 | 0.808 | 119.3 |
| Prox.PhalanxTW | | 0.585 | 0.565 | 0.702 | 0.426 | 2.240 | | 0.752 | 0.728 | 0.834 | 0.436 | 118.7 |
| SonyAIBORobotSurf.1 | | 0.206 | 0.333 | 0.729 | 0.724 | 0.282 | | 0.619 | 0.572 | 0.894 | 0.893 | 102.1 |
| SonyAIBORobotSurf.2 | | 0.589 | 0.468 | 0.885 | 0.876 | 0.560 | | 0.767 | 0.655 | 0.938 | 0.934 | 139.3 |
| SwedishLeaf | | 0.697 | 0.806 | 0.830 | 0.827 | 16.027 | | 0.805 | 0.864 | 0.899 | 0.897 | 441.6 |
| Symbols | | 0.823 | 0.864 | 0.918 | 0.917 | 4.945 | | 0.903 | 0.917 | 0.956 | 0.956 | 1127 |
| ToeSegmentation1 | | 0.221 | 0.181 | 0.737 | 0.728 | 1.136 | | 0.678 | 0.580 | 0.912 | 0.912 | 203.4 |
| ToeSegmentation2 | | 0.486 | 0.379 | 0.862 | 0.809 | 0.780 | | 0.655 | 0.471 | 0.923 | 0.868 | 158.2 |
| TwoPatterns | | 0.908 | 0.870 | 0.965 | 0.964 | 204.000 | | 0.679 | 0.145 | 0.143 | 0.514 | 1719 |
| TwoLeadECG | | 0.484 | 0.438 | 0.848 | 0.846 | 0.742 | | 0.993 | 0.981 | 0.998 | 0.998 | 240.5 |
| Wafer | | 0.977 | 0.936 | 0.996 | 0.991 | 373.891 | | 0.939 | 0.858 | 0.991 | 0.975 | 3329 |
| Wine | | 0.016 | 0.025 | 0.593 | 0.593 | 0.300 | | 0.150 | 0.128 | 0.704 | 0.702 | 62.29 |
| WordSynonyms | | 0.578 | 0.600 | 0.639 | 0.487 | 20.963 | | 0.246 | 0.325 | 0.395 | 0.220 | 764.7 |
| ArrowHead | Ti- | 0.374 | 0.334 | 0.737 | 0.735 | 465.061 | | | | | | |
| Beef | MAE | 0.179 | 0.415 | 0.533 | 0.551 | 483.791 | | | | | | |
| BeetleFly | | 0.219 | 0.344 | 0.750 | 0.733 | 395.046 | | | | | | |
| BirdChicken | | 0.324 | 0.278 | 0.800 | 0.800 | 396.516 | | | | | | |
| Car | | 0.260 | 0.382 | 0.650 | 0.662 | 945.301 | | | | | | |
| ChlorineConc. | | 0.376 | 0.273 | 0.726 | 0.691 | 2665.898 | | | | | | |
| Coffee | | 1.000 | 1.000 | 1.000 | 1.000 | 340.228 | | | | | | |
| DiatomsizeReduction | | 0.980 | 0.971 | 0.990 | 0.984 | 310.567 | | | | | | |
| Dist.Pha.Outln.AgeGrp. | | 0.357 | 0.395 | 0.763 | 0.762 | 1457.101 | | | | | | |
| Dist.Pha.Outln.Correct | | 0.267 | 0.199 | 0.761 | 0.742 | 2509.478 | | | | | | |
| ECG200 | | 0.635 | 0.509 | 0.900 | 0.891 | 584.081 | | | | | | |
| ECGFiveDays | | 0.220 | 0.167 | 0.735 | 0.735 | 402.789 | | | | | | |
| GunPoint | | 0.575 | 0.505 | 0.880 | 0.879 | 472.733 | | | | | | |
| Ham | | 0.130 | 0.102 | 0.686 | 0.685 | 1952.924 | | | | | | |
| Herring | | 0.044 | 0.028 | 0.625 | 0.584 | 1383.155 | | | | | | |
| Lightning2 | | 0.084 | 0.140 | 0.656 | 0.642 | 1512.649 | | | | | | |
| Meat | | 0.622 | 0.646 | 0.867 | 0.832 | 1096.418 | | | | | | |
| Mid.Pha.Outln.AgeGrp. | | 0.066 | 0.032 | 0.571 | 0.416 | 1757.044 | | | | | | |
| Mid.Pha.Outln.Correct | | 0.118 | 0.095 | 0.677 | 0.631 | 2353.275 | | | | | | |
| Mid.PhalanxTW | | 0.442 | 0.409 | 0.604 | 0.437 | 1631.601 | | | | | | |
| MoteStrain | | 0.460 | 0.380 | 0.839 | 0.839 | 322.049 | | | | | | |
| OSULeaf | | 0.216 | 0.290 | 0.483 | 0.479 | 2730.597 | | | | | | |
| Plane | | 0.882 | 0.915 | 0.943 | 0.944 | 562.163 | | | | | | |
| Prox.Pha.Outln.AgeGrp. | | 0.898 | 0.923 | 0.952 | 0.954 | 1407.416 | | | | | | |
| Prox.PhalanxTW | | 0.682 | 0.648 | 0.780 | 0.449 | 1407.906 | | | | | | |
| SonyAIBORobotSurf.1 | | 0.650 | 0.614 | 0.756 | 0.494 | 239.141 | | | | | | |
| SonyAIBORobotSurf.2 | | 0.371 | 0.293 | 0.805 | 0.800 | 274.687 | | | | | | |
| SwedishLeaf | | 0.540 | 0.693 | 0.726 | 0.722 | 2093.867 | | | | | | |
| Symbols | | 0.617 | 0.709 | 0.797 | 0.799 | 456.173 | | | | | | |
| ToeSegmentation1 | | 0.557 | 0.692 | 0.696 | 0.660 | 438.146 | | | | | | |
| ToeSegmentation2 | | 0.343 | 0.215 | 0.815 | 0.740 | 450.744 | | | | | | |
| TwoPatterns | | 0.401 | 0.380 | 0.720 | 0.720 | 4361.683 | | | | | | |
| TwoLeadECG | | 0.359 | 0.356 | 0.696 | 0.698 | 302.257 | | | | | | |
| Wafer | | 0.923 | 0.826 | 0.988 | 0.968 | 5112.925 | | | | | | |
| Wine | | 0.910 | 0.804 | 0.986 | 0.963 | 476.993 | | | | | | |
| WordSynonyms | | 0.509 | 0.503 | 0.563 | 0.394 | 2173.234 | | | | | | |

Table 7: Statistics for selected viral genomes.

| Virus | Strain | Count | Sequence Length | | | |
|---|---|---|---|---|---|---|
| | | | Avg. | Std. | Min. | Max. |
| Influenza-A | H5N1 | 188 | 1368.521 | (21.682) | 1350 | 1457 |
| | H1N1 | 191 | 1421.0 | (15.25) | 1350 | 1468 |
| | H7N9 | 190 | 1403.521 | (12.048) | 1389 | 1444 |
| | H2N2 | 187 | 1430.053 | (17.87) | 1376 | 1467 |
| | H7N3 | 193 | 1423.15 | (21.537) | 1345 | 1468 |
| | **Total** | 949 | 1409.326 | (28.512) | 1345 | 1468 |
| Dengue | DENV-1 | 409 | 10577.812 | (194.4) | 10176 | 10821 |
| | DENV-2 | 409 | 10592.504 | (196.308) | 10173 | 10991 |
| | DENV-3 | 408 | 10614.137 | (132.911) | 10173 | 10810 |
| | DENV-4 | 407 | 10452.469 | (205.208) | 10161 | 10772 |
| | **Total** | 1633 | 10559.328 | (194.74) | 10161 | 10991 |

**Influenza A**   The *Influenza A virus* genome data ($n = 949$) is acquired from the NCBI Influenza Virus Resource (Bao et al., 2008). We consider the genome of segment 6, which encodes the neuraminidase protein, and include sequence samples belonging to subtypes H1N1, H2N2, H5N1, H7N3, and H7N9.

**Denuge**   We consider $n = 1562$ full *Dengue virus* genomes downloaded from the NCBI Virus Variation Resource (Hatcher et al., 2017). We consider all four subtypes of the virus DENV-1, DENV-2, DENV-3, and DENV-4.

We encoded all the sequences as four-channel signals via one-hot encoding, with each nucleotide (A,C,T,G) corresponding to one of the channels. In the case we encounter incompletely specified bases in the nucleic acid sequences (Cornish-Bowden, 1985), we give equiprobable weights to the possible corresponding nucleotides.

**Learning procedure**   Same as the natural language case, we represent the data as a four channel signal and adopt the softmax activation scheme as described in Section 3.2. Since the sequences are of considerable and variable length ($> 1000$ nucleotides, see Table 7), we adopt a stochastic estimation to (8) by randomly sampling length $G = 800$ sub-windows from each of sequences. We take the sample average for each of the sub-window observations similar to (8). We run Algorithm 1 for $N = 1024$ iterations, using the stochastic approximation described above and grid searching across values of $\lambda$. To produce Figures 2b and 2c, we took the learned representations and projected them into two dimensions via UMAP.

## B.8   RUNTIME WITH PARTIAL SVD

In Section 4.2, we report runtime figures where each iteration computes the PROX step via full SVD. In practice, the recovered $\mathbf{B}_\lambda$ matrices often exhibit an exponentially decaying spectrum, allowing singular values to be computed only up to a threshold. In our recovery experiments in Section 4.2, we use all time slices to form the stochastic approximation for Equation (8), and thus, the runtime is primarily determined by the time spent evaluating the VI field.

Table 8 presents the time spent evaluating the VI per outer iteration (each iteration also includes a line search) of the PG Algorithm 1 (in seconds) for four representative UCR datasets and their accompanying statistics. We consider window sizes $d = 20$ and $d = 60$.

In Table 9, we report the spectra of singular values across different values of $\lambda$, and in Table 10, we present the time spent per outer iteration when using either full or partial SVD, as described in the PROJNUC subroutine of Algorithm 1. We note that for small matrices, there is no computational benefit to using the partial SVD. However, as the matrix size increases and for smaller values of $\lambda$, early termination of the singular value search can improve the runtime of the projection step and especially the case when the structure of the data is indeed well approximated by a low rank subspace.

| $d$ | Dataset | $\mathbf{B}$ dimension | Seq. Len. | VI Runtime per iteration (avg. sec) |
|-----|---------|------------------------|-----------|-------------------------------------|
| 20 | Meat | $82 \times 120$ | 448 | 4.00 |
| | ECG200 | $82 \times 200$ | 96 | 1.12 |
| | DistalPhalanxOutlineAgeGroup | $82 \times 539$ | 80 | 2.83 |
| | Wafer | $82 \times 7164$ | 152 | 78.31 |
| 60 | Meat | $242 \times 120$ | 448 | 5.57 |
| | ECG200 | $242 \times 200$ | 96 | 0.99 |
| | DistalPhalanxOutlineAgeGroup | $242 \times 539$ | 80 | 1.90 |
| | Wafer | $242 \times 7164$ | 152 | 103.53 |

Table 8: Statistics for four representative datasets. The time to evaluate the VI field given in Equation (8) is given in seconds per outer iteration of Algorithm 1 across all choices of $\lambda$ given in Table 10.

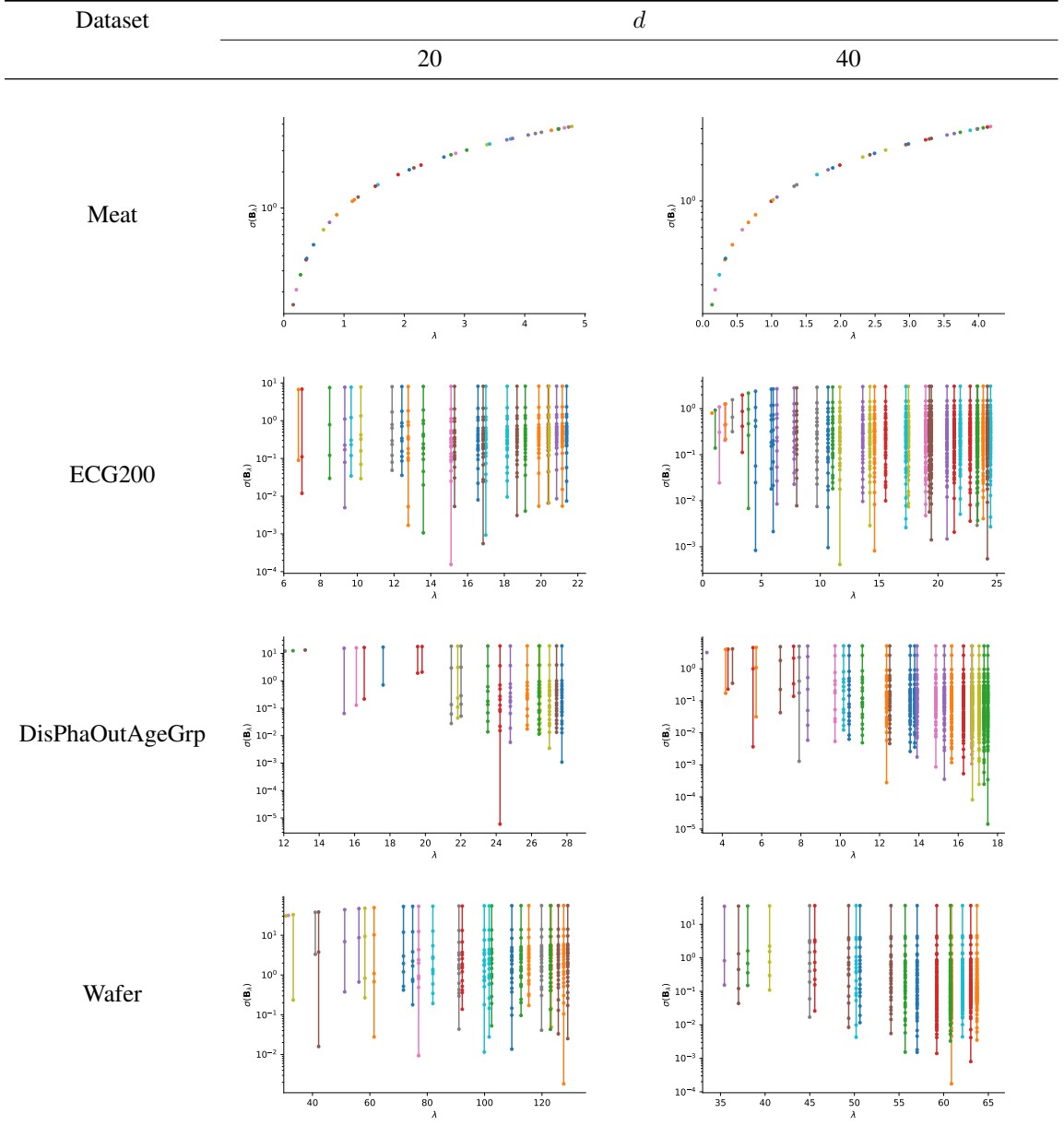

Table 9: Spectrum of recovered singular values for $\mathbf{B}_\lambda$ with differing levels of $\lambda$ for different real world datasets from UCR data as described in Table 8. When $\lambda$ is smaller, it is generally sufficient to compute a fewer number of singular values.

| Dataset | $d$ | |
| --- | --- | --- |
| | 20 | 40 |

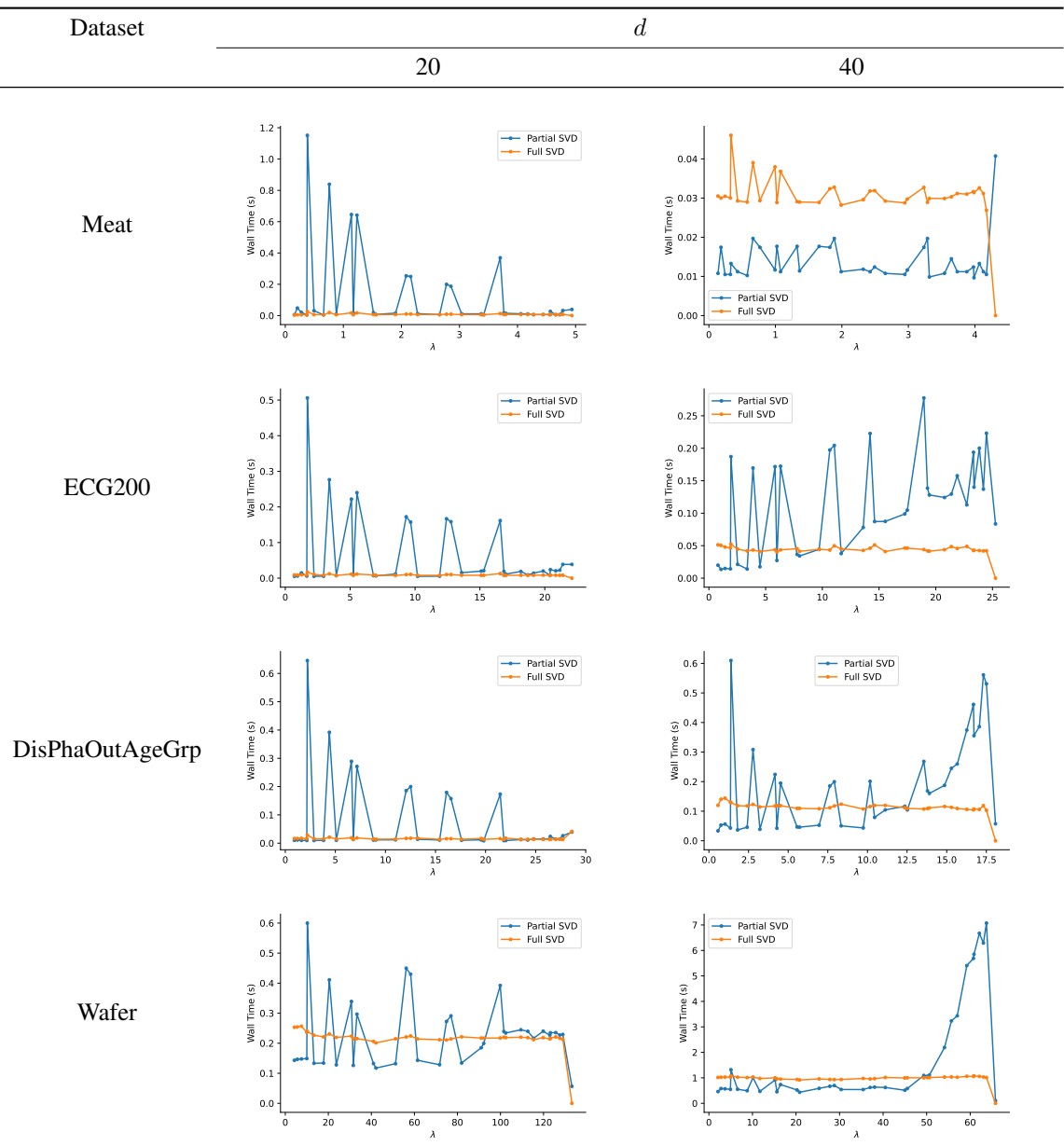

Table 10: Average outer iteration runtime (seconds) of Algorithm 1 using PROXNUC for the PROX step (computing incremental singular values) versus using the full SVD across different $\lambda$ values for datasets in Table 8.

