# OpenReview forum: "Nonlinear Sequence Embedding by Monotone Variational Inequality"
_ICLR.cc/2025/Conference — ICLR 2025 Spotlight_

### Official Review · Reviewer_iV1B · 2024-10-30

**Soundness:** 3
**Presentation:** 2
**Contribution:** 2
**Rating:** 8
**Confidence:** 3

**Summary:**

The paper introduces an unsupervised representation learning framework for sequential data, formulated as a low-rank matrix recovery problem using a convex optimization approach with monotone VIs.
The framework assumes a shared underlying domain among sequences and enforces this assumption through a low-rank constraint on the learned representations, implemented as a constraint of a nuclear norm ball.
This method is theoretically grounded and shows competitive empirical performance, while achieving provable recovery guarantees.

**Strengths:**

1. **Theoretical Rigor**: The proposed framework is strongly backed by theoretical foundations, leveraging low-rank matrix recovery and convex optimization over monotone VIs.
The approach is both well-formulated and analytically justified, with provable recovery guarantees under the low-rank assumption, which strengthens the theoretical contribution.
2. **Empirical Performance**: Experimental results support the framework’s ability to recover parameter matrix and demonstrate competitive performance across real-world data.

**Weaknesses:**

1. **Benchmark Comparisons**: Given the objective of learning representations, it would strengthen the work to benchmark against established representation learning techniques like contrastive learning and masked modeling.
2. **Minor Issues**
    - **Clarity**: Consistency in notation, particularly in equations and indexing conventions such as the program notation in eqs. (6-7) and the loss function in eq. (14), would enhance readability.
    - **Notation**: The notation should be updated to $\mathcal{H}_{i,t}$ for clarity and consistency.
    - **Errata**: There are minor typos, such as repeated words on pages 1–2 and “spacewof” instead of “space of” on p6.

**Questions:**

1. Could the authors elaborate on their choice not to include benchmarks against popular representation learning methods?

---

> ### Author Response · Authors · 2024-11-24
>
> *TS2Vec* is a quite popular *contrastive learning* based method for time-series representation learning included in our unsupervised time-series classification task. Based on reviewer feedback, we also added a comparison with a representation learning method based on *masked modeling* for time series data (we picked Ti-MAE). (We also tried another masked modeling approach SimMTM, but the representation quality was not on par with Ti-MAE and TS2Vec, so we excluded it).
>
> We have included also illustrations of the representation projected into 2D in Appendix B5, which compares what is recovered by our method vs TS2Vec (contrastive learning) and Ti-MAE (masked modeling)
>
> If the reviewer has additional suggestions for other methods, we are open to incorporating the evaluation.
>
> Apart from that, we do want to briefly emphasize the setting of the problem, which influences our methodology:
>
> We work with a collection of (partially) observed sequential data (e.g., time-series, language data, count processes).  The sequences which we observe all come from a common domain, but that domain may vary widely (e.g., could be sequence data, or it could be a vector time-series). The observations of each sequence are limited, and their dynamics may be quite different from each other (e.g., healthy vs sick patient). We must rely on limited, noisy, heterogeneous data. In this setting, we feel it is important to highlight the matrix sensing perspective as another direction for sample efficient representation learning beyond the traditional contrastive learning (Energy Based Model) or masked modeling perspective.
>
> Errata:
> Thanks for pointing these out — Updated in new version of manuscript
> Eqns: (6-7), (14). Updated the indexing convention to be more consistent, and made the dependence on only the history of the particular sequence $\mathcal{H}_{i,t}$ explicit.

---

> > ### Comment · Reviewer_iV1B · 2024-11-25
> >
> > Thank you for addressing my concerns and for incorporating masked modeling as an additional comparison.
> > The inclusion of representation illustrations is particularly helpful for clarifying the distinctions between your method and existing approaches.
> >
> > I also appreciate your explanation of the unique challenges in your problem setting and your rationale for framing the matrix sensing perspective as a sample-efficient alternative.
> > This context highlights the significance of your contribution to advancing representation learning under noisy and heterogeneous conditions.
> >
> > With these thoughtful revisions, my concerns have been effectively addressed, and I will update my evaluation accordingly.
> > Thank you for your detailed and constructive response.

---

### Official Review · Reviewer_7R4f · 2024-11-02

**Soundness:** 3
**Presentation:** 3
**Contribution:** 3
**Rating:** 8
**Confidence:** 3

**Summary:**

The paper introduces a novel approach for unsupervised learning of low-dimensional representations for multiple related time series using a monotone variational inequality framework. The approach combines traditional autoregressive modeling with a low-rank regularization, which leads to efficient matrix recovery and nonlinear sequence embedding while maintaining convexity. The authors test their approach through model recovery experiments on synthetic linear AR sequences and applications to real world time series classification and more complex symbolic datasets, such as language and genomics.

**Strengths:**

- The paper is neatly written, and I overall found the presentation of the figures and clarity of writing nice.
- The paper looks mathematically rigorous and well-rooted in related literature and approaches.

**Weaknesses:**

- Method Section: I am admittedly not familiar with much of the literature cited in this section. I appreciate the mathematical rigor but found some details in this section hard to follow. This also makes it harder to assess which parts of this section are novel contributions. A summary section at the end of the introduction or the method section that explicitly states what is novel about this approach could make the main contributions of the paper clearer.

- Results: Linear and Nonlinear Models: The shift between linear and nonlinear models was somewhat confusing on the first read.
Section 4.3 mentions learning representations for sequences with nonlinear dynamics, but the nonlinear approach seems to be applied earlier in Section 4.2 as well.

- The model recovery experiments are specifically for the linear case. Clarifying why only the linear model was used for parameter recovery while not applied to real data would help. Is the idea to test the nuclear ball regularization separately in a simpler setting?

- On the other hand, including a direct comparison between the linear and nonlinear models in terms of performance on the real-world data might also be interesting. Vice versa, it could also be beneficial to add parameter recovery experiments for the nonlinear case as well.

- The role of  $\lambda$: what role does the low-rank constraint play on performance? You mentioned you tuned lambda on the training set. Can we learn something about the complexity of the datasets depending on which lambda is optimal? How much better does performance get using $\lambda$, compared to the unconstrained case?

- Choice of comparisons: None of the evaluated comparisons incorporate low-rank constraints. While as the authors mentioned, time series classification is just one example downstream task (that is easy to quantify), some more direct comparison to other low-rank approaches could make the specific advantages of your method more clear.

- Limitations:
There is no discussion of limitations. Besides descriptions of the approach/metrics, it is hard for me to assess what potential shortcomings/challenges the approach faces if I were to use it in practice. Tuning for $\lambda$ is discussed in a lot of detail, but not so much for other aspects of the algorithm. This part could also include something on potential extensions, such as handling higher-dimensional examples.

**Questions:**

- Related Work: this is a broad field, so it’s challenging to cover every related approach. A class of approaches that goes in a similar direction which could be mentioned as well are hierarchical (Bayesian) time series models, e.g. Random Effects Models for Longitudinal Data (Laird and Ware, 1982) or dynamical hierarchical models (Zoeter and Heskes, 2003). Likewise, in the field of dynamical systems modeling, there are some approaches going in a similar direction, extracting low-rank representations across multiple observed time series (e.g. CoDA, Kirchmayer, ICML 2022)

Typos:
- indicates twice around page break between pages 1 to 2
- Page 3: sufficiently captured an order (missing “by”)
- Page 6: spacewof matrices

---

> ### Author Response · Authors · 2024-11-24
>
> We thank the reviewer for their thorough feedback. To address their questions and concerns:
>
> ### Development of Method Section:
>  We have updated the discussion at the end of Section 2, just before the Methods section, to explicitly outline the development (and highlight what the new ideas are). This aims to provide an accessible setup for the Methods section. We also updated the related work section to highlight what part is existing and what part is new.
>
> To address the novelty directly: many ideas used in this paper have been floating around for some time from two streams of literature: (1) *Matrix Sensing*, and (2) *Statistical Inference via Convex Optimization*. **The idea to treat representation learning across sequences/time-series data as a tractable convex “low-rank matrix sensing” problem is new.** We feel that this perspective holds promise, and connects, from first principles, those three areas — time-series/sequence representation learning, statistical inference via convex optimization, and matrix sensing — that have not been explicitly connected in this way before, and this **perspective is especially useful for limited/partially/highly heterogenous observed data**.
>
> The exact observation model in Equation (1) is rooted in the literature on *Statistical Inference via Convex Optimization* (see book of the same name, Juditsky and Nemirovski 2020), which addresses the most general setting of solving inverse problems through monotone operators. Our contribution extends and specializes this to time-series learning — we provide the methodological and algorithmic tools for this connection. The specific algorithmic solution we provide for solving the monotone VI problem leverage standard first-order methods for convex optimization, with problem-specific adaptations (e.g., addressing the nuclear ball setting with monotone VI and adaptive estimation of modulus of convexity), the various tools to address these computational problems are known in mathematical optimization community, but may not be as well known to the broader machine learning community. Work was required to make them useful to our setting. Likewise the ideas from matrix sensing have been developed and floating around in a number of other settings (classic examples are recommender systems, and MRI image reconstruction), but to establish connection to recover representations of time-series/sequence collections and to apply it in this class of statistical inference problem is a new contribution.

---

> > ### Author Response · Authors · 2024-11-24
> >
> > ### Results: Linear and Nonlinear Models and Design of Experiments
> > - We also have experimental results for synthetic sequence data generated from the state evolution of random Markov chains. The parameter recovery results are essentially the same as the linear case. Because they are so similar, we cut them so as to not distract from the main message of the section — that low rank constraint can lead to better recovery of true parameters and higher quality latent space at very little cost to the “autoregressive” performance of the model (as measured by the least squares loss). The reason why we exposit the linear rather than the sequence recovery is simply because it is easier to evaluate and explain the reconstruction error in the linear case, and the least squares loss is easier to exposit as compared to have to explain the meaning of the magnitude of the variational inequality field (the corresponding measure in the nonlinear case).
> >     - *We have now included a brief discussion and some figures from these experiments in the appendix .*
> > - As for real data experiments. In our context, there are two ways to introduce nonlinearity to the model (A) modify the link function or (B) augment the input signal by various nonlinearities (e.g., we use the time-series themselves, as well as their first finite differences)in our real data experiment). Varying (A) is primarily to model different phenomena, e.g., continuous time sequences vs sequences where the output domain should be limited to a probability vector, which is why we separate the discussion in two parts — these are distinct tasks. We note that one may also pick (A) from a convex set, at essentially no cost to the optimization.
> > - On this point, we want to highlight some important observations. In the data setting which we experiment with (medium sized, but from a broad variety of biological, social, and industrial contexts), we can use an easily interpretable and relatively simple observation model (vector autoregressive through a arbitrary monotone link function of Equation (1), coupled with this low rank constraint to get good performance in unsupervised time series classification task (as compared to neural network model based on contrastive learning), without much tuning. Though the time-series representation learning literature is vast, we have not found a comparable method which relies on *low-rank representation* in the same/similar sense to us (e.g., not in the sense of there simply being latent factors to learn, or "low rank adaptation" in the case of sequence data).
> >
> > ### Role of $\lambda$
> > The role of the nuclear penalty term $\lambda$ is to balance the emphasis on the individual sequence characteristics, versus the information known from the other sequences (domain learning). The basic intuition is that with unconstrained $\lambda$, you simply fit a nonlinear autoregressive model — Equation (1) — to each sequence individually. This can give you as good performance as possible using the available data of each sequence (which is exactly the observations you have of that focal sequence). As you make the rank smaller, you allow for the information from the other sequences to indirectly inform (through the tightening the parameter subspace) your current prediction. This results in improved recovery altogether. However, you do trade-off the number of factors which govern each sequence (so you gradually lose expressivity — in the extreme when the signal is rank one, the parameters of each model can only be multiples of each other), eventually this becomes the dominant factor, so the performance goes back up. This intuition is supported by theory from the low-rank matrix sensing literature, where adding in penalties in the $\ell_1$ (or, in this case, inducing sparsity in the singular values) leads to improved signal recovery, when the underlying structure (referring to the domain where we are drawing the generator/observation pairs from) is low rank. You must pick $\lambda$ to be sufficiently small, as if not, the output representation will not be low rank.
> >
> > ### Related Work:
> > We chose to exposit the literature primarily in the area of convex optimization and matrix sensing. The references brought up are relevant and useful, we have included them in the literature section.

---

> > > ### Author Response · Authors · 2024-11-24
> > >
> > > ### Limitations:
> > > The primary limitation is the assumption of the model: which (A) your original data must be well modeled by an vector autoregressive processes through some monotone link function (Equation (1)), though we have made an effort to make the setting as broad as possible using monotone VI to overcome non-convexity, there is no guarantee of universal approximation. The most fundamental reason why, at present, we leave the assumption of the true observation model to be convex, is to ensure the identifiability of the model parameters during the optimization procedure, as the second assumption (B) is that the parameter space of the observed samples lie on a low dimensional linear subspace of the entire possible parameter space. This is dependent in turn on the choice of link function and the data topology. Typically we require that (A) and (B) both be satisfied, in the cases which this is not true, our model does not perform well (e.g., there are some degenerate cases). However, empirical experiments point to the assumptions being true for a lot of data from a lot of different domains. A natural next step is to relax these assumptions, e.g., to make our model more expressive in modeling each sequence, as well better handle high dimensional time series data, without sacrificing the efficiency of our method to work with small but highly heterogeneous data samples.
> > >
> > > As to why $\lambda$ is discussed, in contrast to other hyperparameters is because most of the other hyperparameters can be automatically tuned (discussed in the algorithmic section of the appendix, and which requires some machinery from constrained first order convex optimization).
> > >
> > > ### Errata:
> > > Thanks for pointing these out. Fixed in updated manuscript.

---

> > > ### Comment · Reviewer_7R4f · 2024-11-25
> > >
> > > Thanks for the detailed and lucid reply, and the rewrite and novel results! I am happy to join the overall positive assessment of the paper by the other referees and vote for acceptance.

---

### Official Review · Reviewer_9iKd · 2024-11-04

**Soundness:** 3
**Presentation:** 3
**Contribution:** 3
**Rating:** 6
**Confidence:** 2

**Summary:**

This work introduces an unsupervised method for learning low-dimensional representations of sequences. Specifically, it assumes that each sequence relates through a low-rank latent autoregressive matrix, and, by casting the problem as matrix recovery, the authors derive an efficient algorithm to obtain the latent matrix. The authors evaluated the resulting representations obtained by the proposed method in the UCR time-series classification task and clustering on language and genome sequences.

**Strengths:**

- The paper is well-written and easy to follow.
- Bringing the matrix recovery technique in sequential learning enables us to efficiently compute latent autoregressive parameters with some theoretical guarantees such as computational complexity and parameter recovery guarantees, which look a good contribution to the representation learning of sequential data.

**Weaknesses:**

- Regarding the clustering experiments; it is hard to tell the model's significance since the authors do not include any other model's results. I see the obtained representations by the proposed method are well-clustered in every dataset, but how do they become when using different models (e.g. models shown in table 2.)?

**Questions:**

Typo
- l:285 the spacewof -> the space of ?

---

> ### Author Response · Authors · 2024-11-24
>
> We thank the reviewer for their feedback.
>
> To answer your questions on clustering experiments:
>
> Assuming that review refers to those clusterings shown in Figure 2, which deal with symbolic sequences (book vs paper paragraphs, viral genomes).
>
> As for the time-series data, we have now included similar illustrations to Figure 2, for those methods which output a vector representation for each time-series (e.g., the contrastive learning approach TS2Vec, and now also a masked modeling approach Ti-MAE) to the Appendix, as to complement the discussion existing in Section 4.2. We hope this helps to highlight the cases both where we learn an effective representation and where our method falls short (the limitation). Our method, for the most part (there are certain patterns which our model captures, and the other two do not, and vice versa), performs similarly to those methods, while being faster (as the underlying problem is convex) and more sample efficient when there are limited observations. One main aim of our work is to present an alternative perspective for sequence/time-series representation learning beyond those methods, which is grounded in the matrix-sensing perspective (which we hope is useful) as adapted to time series data, and lets us deal with the balance between the sequence individuality and the global domain of sequences.
>
> The symbolic sequences are presented for the matter that we wished to highlight the extension of treatment to the sequence case. In the natural language and genetic settings, particularly, one may learn something of essentially similar quality using the embedding from the foundation model (the tradeoff being that one would need to train one of these — not possible in many data domains), we want to highlight that one does not need to use a large pre-trained model to do this, especially in the setting of starting from scratch, with limited data, which is common with many sequential data from specialized domains.
>
> Errata:
> Thanks for pointing out the typo — fixed in the updated version of the manuscript.

---

> > ### Comment · Reviewer_9iKd · 2024-11-24
> >
> > Thank you for the response! I see that my concern has been addressed, and I will keep my existing score.

---

### Official Review · Reviewer_wK5W · 2024-11-04

**Soundness:** 3
**Presentation:** 3
**Contribution:** 3
**Rating:** 8
**Confidence:** 3

**Summary:**

This work aims to learn the low dimensional representation of sequential data with nonlinear dynamics. It combines low-rank matrix recovery and monotone VI to learn low-rank representation of nonlinear signals, while maintaining the signal recovery problem’s convexity for acceleration. The low-rank constraint is enforced through nuclear norm regularization with proximal algorithms, while the modeling of the nonlinear dynamics and the monotone VI both rely on the monotone nonlinear link function.

**Strengths:**

1. Well written work, with comprehensive introduction, clear problem statement, detailed methodology presentation and thorough experiments. Pseudocode is also given.
2. Dimension reduction or representation of sequential data is indeed a crucial problem.
3. The method looks novel and sound.
4. The experiments can clearly reflect this method’s efficacy.

**Weaknesses:**

## Major

1. I think the performances of linear and nonlinear monotone autoregressive models (both with low-rank regularization) might need to be compared with further experiments. It seems to me that the choice of the link function $\eta$ is still kinda restrictive and problem specific, as it needs to be monotone and remains fixed after being selected. Intuitively if a nonlinear $\eta$ is very similar to $\eta(\mathbf{x}) = \mathbf{x}$, the models using them may not be significantly different in terms of performance. Not sure if spending so much effort making the autoregressive model a tad nonlinear is worth it, especially considering that the nonlinear one needs the additional evaluation of $\mathcal{A}$ and $\mathcal{A}^*$.
    - However, l like this paper overall, so this is not fatal.
    - See Question 2 for more detail.
2. The authors had better mention their method’s limitations and suggests some future research directions. For instance, a few things mentioned in the my questions.

## Minor

1. You should briefly explain why the *monotone property* of the link function is important when you first introduce it on Line 122 and 132 (like simply referring to Sec. 3.2), or cite related works given that you have so many examples.
In addition, you should explicitly refer the readers to Line 263 for the accurate definition of  “monotone”. A better option is to turn Line 262 to 269 into a formal Definition block with number (given how crucial monotonicity is to your method), then \ref it on Line 122.
2. Line 136, is the subspace linear? If so, you had better mention it explicitly here, or claim that all subspaces in this work mean linear subspaces.
3. “Geometry” in phrases like “geometry of the subspace” “geometry for the entire domain” “geometry of the representations” is a very awkward term for your work. The subspaces in this work are just flat planes with no special “geometry”. It’s not like you have to deal with Riemannian metric, angle, geodesic, curvature and so forth on a nonlinear submanifold.
    1. For instance, on Line 154, you can basically just say “the subspace…captures…’’
    2. If you insist on using “geometry”, be clear about what you mean by it.

## Trivial:

1. Use the correct citation format: \citep and \citet. For example, Line 089.
2. Typos. For instance:
    1. Line 053: duplicate “indicates”
    2. Line 106 “thewfirst”; Line 285 “spacewof”
    3. Line 115, missing “by”
3. Clarify your special notation when it’s introduced. What is “vec()”? Vectorize? Flatten?
4. In Table 2, make the best Avg. in each column stand out with bold font.

**Questions:**

1. Does the link function $\eta$ have to be continuous?
    - Just curious, where does the name “link” come from originally? Why is it called "link"?
2. Because the $\eta$ is monotone and fixed, does the model in (1) in general have enough complexity to capture the nonlinear dynamics of a complicated sequence, e.g., a sequence with many high frequency components with rather large amplitude?
    - Which hyperparameters of it can affect the complexity?
        - If my sequential data have very low $C$ and complicated dynamics, is increasing $d$ basically the only way to increase your model's complexity? Can increasing $d$ always improve the signal recovery effectively?
    - I guess that if the link function is not good enough, $\mathrm{rank}~\mathbf{B}$ cannot be sufficiently reduced, right? How to choose or design one?
3. It seems like the model can be used for denoising by limiting $\mathrm{rank}~\mathbf{B}$. What if some very important signal components have very low SNR, and I don’t care much about the principal signal components with large SNR? Can the model potentially differentiate between them and only remove the noise?
4. Why does the relative reconstruction error have a quadratic relation w.r.t. $\lambda$ in Fig 1.a? Underfitting and overfitting?
5. How long does it take to search for $\lambda$? Does it scale with the problem’s dimensionality?
6. What is the speed difference between recurrence (8) and (15)?
7. How widely applicable is the assumption "we suppose..." on Line 114 in reality?

---

> ### Author Response · Authors · 2024-11-24
>
> We thank the reviewer for their feedback. To answer your questions:
>
> ### Questions:
> 1. The link function $\eta$ does not need to be continuous, only strongly monotone. The term “link function” originates from literature on generalized linear models (Nelder and Wedderburn, 1972), describing the connection between the predictor and response variable. The observation model in Equation (1) generalizes the generalized linear model framework.
> 2. There are cases where the parametric model in Equation (1) may lack sufficient complexity to effectively capture certain observed phenomena. We will first address these cases (and how to mitigate them) and then explain our focus on the vector autoregression with a monotone link setting.
>     - Sequences with many high-frequency components and large amplitudes (e.g., EKG signals, included in our testing set) are generally not a problem. However, some real-world datasets in the UCR benchmark present challenges where our model underperforms, particularly in the unsupervised time-series clustering task.
>    - The performance of the model, at least on the UCR benchmark (which includes diverse time-series data across social and natural sciences), is comparable to the neural representation model Time2Vec.
>    - We chose the model in (Equation (1) instead of a more general neural network) because it is as expressive as possible while preserving convexity. The convexity of the model is not only necessary establish convergence and parameter recovery guarantee of the model, but more fundamentally to ensure its identifiability, and uphold the hypothesis that the generating parameters for each sequence may lie in a low-rank space—a property not guaranteed with non-convex models like general neural networks due to their optimization landscape.
>    - That said, this reliance on the convex model in Equation (1) is a limitation, as it does not have universal approximation guarantees.
> The main objective of the link function is generally to constrain the “type of the problem”, e.g., one picks SoftMax to ensure that each channel represents a probability across a number of different symbols/objects, or you pick sigmoid to represent a binary (Bernoulli) process (e.g., whether a neuron has fired),
>      - To relax the choice of $\eta$ one may also pick the link function $\eta$ from a convex set, instead of setting it beforehand.
> One can also add additional (duplicate or different) channels to your data, this can improve the performance. Which addresses the possibility of having low $C$ and complicated dynamics. The cost comes in with the fact you will need to pay for additional channels.
> To relax $\eta$, it can be chosen from a convex set rather than predefining it. Another approach to enhance performance is adding additional or feature-engineered channels to the data. This method, which addresses cases with low $C$ and complex dynamics, incurs a cost in increased dimensionality. For example, in UCR recovery experiments, we used both the raw data and its first finite difference to introduce nonlinearity and improve performance
>      - Increasing $d$ (dimensionality of the past data) does not always improve signal recovery, as distant past data points may be less relevant for predicting the next state.
>      - If the link is not enough to help capture the dynamics, then the rank(B) may be high. It may be possible to remedy by picking a better $\eta$, or by the tricks above, or it may well be that the data cannot be modeled in this fashion (e.g., it is far too complex, in which case it is difficult to say what the representation means in the first place). This is another limitation of our work.
>      - That said, without too much tuning our method seems to work well in a lot of cases and is backed up by the matrix sensing theory, and is at least comparable with faster runtime compared to neural network-based methods, which rely on contrastive/masked modeling in the data limited regime we deal with. Our method performs better when there are limited samples for which the sample efficiency of neural networks comes at a detriment.
> 3. Yes, though not necessarily in the sense of “denoising” the observed signal (e.g., like a high-pass filter). We note that the “signal recovery“ we are looking at is in the sense of (approximately) recovering the “true generating dynamic” which underlies the observed observations. Such a dynamic may result in a number of different sequences, (e.g., random noise, depending on the original seed observations). The “denoising” in this sense, would be that by constraining the sequence parameters to lie in a low rank space, this will eliminate much of the noise in the space of model parameters, which would help augment the problem of low signal-to-noise ratio, due to the fact we only have limited observation of each sequence.

---

> > ### Author Response · Authors · 2024-11-24
> >
> > 4. The basic intuition for the unimodality is that with unconstrained $\lambda$, you simply fit a nonlinear autoregressive model — Equation (1) — to each sequence individually. This can give you as good performance as possible using the available data of each sequence (which is exactly the observations you have). As you make the rank smaller, you allow for the information from the other sequences to indirectly inform (through tightening the parameter subspace) your current prediction. This results in improved recovery altogether. However, you do trade-off the number of factors which govern each sequence (so you gradually lose expressivity — in the extreme when the signal is rank one, the parameters of each model can only be multiples of each other), eventually this becomes the dominant factor, so the performance goes back up. This intuition is supported by the theory from the low-rank/compressive matrix sensing literature
> > 5. Because of unimodal structure as discussed (Qn. 4), search for $\lambda$ does not scale with the problem dimensionality, you can converge to the optimal $\lambda$ in solving $\epsilon \leq k 2^{-t}$ in $t$ versions of the program.
> > 6. The only additional cost is applying the link function to the input at each time. The convergence properties remain the same in both the linear and nonlinear case. This is because the recurrence of (8) is a special case of (15). If one sets as the $\eta$ the identity function, then the recurrence is exactly the same as (8) — in fact the vector field of the monotone VI is exactly the gradient field of the least squares loss (true $\eta = \mathbb{Id}$, not true in general)
> > 7. See discussion for (Qn 2.)
> >
> > ### Next Steps and Limitations:
> > - See the discussion of (Qn 2., and the discussion of weaknesses) Updated to include some next steps/weaknesses included in the conclusion section. T
> >
> > To summarize:
> > - Not all phenomena are well modeled by the scheme in Equation (1), and not all phenomena can be well captured in low rank space (and this may be dependent on clever choices of $\eta$ or different ways to augment the signal)
> > - The model is formulated as it is because it is the most general it can be while still maintaining the problem convexity (needed for as to ensure identifiability and regularity of the parameter space, when we impose the low rank constraint).
> > - How to extend the general observation model, where the generating dynamics are drawn from common domain (i.i.d.), and then realized via some random processes evolving according to the dynamic for which we receive (partial obviation),  where a recovery process can handle cases of universal approximation in a (1) identifiable and (2) sample efficient manner is an natural next step.
> > - Some extensions: How to perform the procedure in an online fashion? Nonlinear subspace learning (though we believe it better to introduce the nonlinearity in other places, e.g., with the formulation of Equation (1), e.g., by some kind of further relaxation, or introduce a neural network as the parameterization)?
> >
> > ### Minor and Errata: typesetting
> > - Yes, a linear subspace of the parameter space. Fair enough, we have updated to use the term “linear subspace”
> >    - Nonlinear manifold is an interesting direction, though the needed extensions to the present available theory in low-rank estimation to adapt to general manifold structure is not clear yet.
> >
> > Thanks for pointing the errata.
> > - $vec()$ means flatten (arrange the entries in arguments into a single column vector), added one clause to explain this.

---

> > > ### Comment · Reviewer_wK5W · 2024-11-25
> > >
> > > Thanks for your detailed answer. I will maintain my score.

---

### Author Response · Authors · 2024-11-24

We wanted to thank the reviewers for their thorough feedback.

Based on their suggestions, we have revised the introduction, problem statement, and conclusion to better contextualize our work and discuss the strengths and limitations of our method. We have also added some *additional experiments and illustrations*, including:
1. A comparison to a masked modeling approach (Ti-MAE), complementing our existing comparison to TS2Vec, a contrastive learning method.
2. An illustration of synthetic nonlinear sequence recovery (from Hidden Markov Models).
3. Projections of the learned representational space for real data, comparing the quality of representations generated by our model versus contrastive learning and masked modeling.

**We have highlighted those updates in blue in the manuscript**. We also wanted to thank the reviewers for pointing out typographical, notational, and grammatical errata.

As a final note, we wanted to *provide some perspective on this work*.

While low-rank matrix recovery has been studied extensively, its explicit connection to learning representations across heterogeneous time-series (from common domain) is new. We feel this connection contrasts from the mainstream representation learning perspective, and is an important conceptual and mathematical framework by which we can understand the representation learning problem, especially in settings with partial or limited observations. In particular it provides a method by which we can understand the balance between the learning of domain information, versus unique features of each sequence. These ideas have been touched in other papers before (e.g., some of the masked modeling papers touch on learning a manifold of time-series) but are not formalized and not analyzed in the same manner as us.

In this work we frame the task as a rank-constrained convex parameter recovery problem. This approach is particularly amenable to settings with partial or limited observations, allowing similar sequences to inform the representation of a focal sequence. This observation model is both a strength and limitation: on one hand we have made it *as general as possible while still maintaining convexity*, and os thus flexible enough to handle a number of differnt scenarios — notably probabilistic modeling of symbolic data —, is sample efficient, and demonstrates empirical performance comparable to methods based on contrastive learning and masked modeling. On the other hand, reliance on convexity to ensure regularity and identifiability limits its ability to provide universal approximation guarantees. Our method performs well under low-rank and monotonicity assumptions, is sample-efficient, and is faster in limited-data settings, as shown in most cases. One limitation is that the performance declines when these assumptions are violated, which can be seen in certain UCR datasets, where it may be outperformed by energy-based approaches, particularly where in very data-rich scenarios. Future work could explore alternative objectives within the VI framework and various non-convex extensions to address these limitations.

---

### Meta-Review · Area_Chair_Tnhv · 2024-12-07

**Metareview:**

The paper develops an unsupervised method for learning low-dimensional representations for time-series data, i.e., sequences. The developed method is under the assumptions that (1) each sequences arise from a common domain, following its own autoregressive model and (2) these models are related through low-rank regularization. The problem of finding low-dimensional representation can therefore be reframed from the lens of matrix recovery. The authors derived an efficient algorithm to recover the latent matrix and evaluated the recovered representation in UCR time-series classification task, clustering on language and genomic sequences.

Overall, most reviewers appreciate the strengths in theoretical rigor and empirical results of this paper. Most questions raised in the original reviews are for clarification and suggestions for editorial fixes such as notations and further intuition/discussion. The authors have addressed those well in their detailed rebuttal.

All reviewers vote for acceptance. This is a clear case where the contribution is well-recognized by the entire review panel.
I therefore recommend acceptance for this paper.

**Additional Comments On Reviewer Discussion:**

The reviewers have been positive about this paper. The rebuttal of the authors also help retain the positive ratings.

---

### Decision · Program_Chairs · 2025-01-22

Accept (Spotlight)